# Bi-directional regulation of cognitive control by distinct prefrontal cortical output neurons to thalamus and striatum

Sybren F. de Kloet [1,4✉], Bastiaan Bruinsma[1,4], Huub Terra[1,4], Tim S. Heistek[1], Emma M. J. Passchier[1,2], Alexandra R. van den Berg[1], Antonio Luchicchi[3], Rogier Min [1,2], Tommy Pattij[3] & Huibert D. Mansvelder [1✉]

The medial prefrontal cortex (mPFC) steers goal-directed actions and withholds inappropriate behavior. Dorsal and ventral mPFC (dmPFC/vmPFC) circuits have distinct roles in cognitive control, but underlying mechanisms are poorly understood. Here we use neuroanatomical tracing techniques, in vitro electrophysiology, chemogenetics and fiber photometry in rats engaged in a 5-choice serial reaction time task to characterize dmPFC and vmPFC outputs to distinct thalamic and striatal subdomains. We identify four spatially segregated projection neuron populations in the mPFC. Using fiber photometry we show that these projections distinctly encode behavior. Postsynaptic striatal and thalamic neurons differentially process synaptic inputs from dmPFC and vmPFC, highlighting mechanisms that potentially amplify distinct pathways underlying cognitive control of behavior. Chemogenetic silencing of dmPFC and vmPFC projections to lateral and medial mediodorsal thalamus subregions oppositely regulate cognitive control. In addition, dmPFC neurons projecting to striatum and thalamus divergently regulate cognitive control. Collectively, we show that mPFC output pathways targeting anatomically and functionally distinct striatal and thalamic subregions encode bi-directional command of cognitive control.

[1] Department of Integrative Neurophysiology, Center for Neurogenomics and Cognitive Research, Vrije Universiteit Amsterdam, Amsterdam Neuroscience, Amsterdam, The Netherlands. [2] Department of Child Neurology, Emma Children's Hospital, Amsterdam University Medical Centers, Vrije Universiteit Amsterdam and Amsterdam Neuroscience, Amsterdam, The Netherlands. [3] Department of Anatomy and Neurosciences, Amsterdam University Medical Centers, Vrije Universiteit Amsterdam and Amsterdam Neuroscience, Amsterdam, The Netherlands. [4] These authors contributed equally: Sybren F. de Kloet, Bastiaan Bruinsma, Huub Terra. ✉email: s.f.de.kloet@vu.nl; h.d.mansvelder@vu.nl

Cognitive control involves the ability to suppress undesirable actions and remain attentive to relevant stimuli. The medial prefrontal cortex (mPFC) is highly involved in these processes, as shown in the lesion, pharmacological, optogenetic, and chemogenetic experiments[1–4]. Distinct neuronal activation patterns across mPFC subregions, cell types, and behavioral subdomains often underlie attention and inhibitory control[3,5,6]. However, there is substantial heterogeneity in timing, location, and origin of brain activity associated with behavior[7]. For instance, the dorsomedial PFC (dmPFC; defined here as premotor, dorsal prelimbic, and anterior cingulate cortex) has been associated with longer windows of activity than the ventral mPFC (vmPFC; as the infralimbic and ventral prelimbic cortex) during delay periods in cognitive control tasks[4]. Neurons in the dmPFC and vmPFC can further be classified based on their projection target and transcriptomic profile[8–10]. Functional studies have established a role for projection-specific mPFC populations in goal-directed behavior[11,12]. This suggests that studying the function of projection-specific neurons may lead to a better understanding of the role of specific neural populations and circuits in attention and inhibitory control.

Several downstream targets of the mPFC are associated with attention and inhibitory control. The mediodorsal thalamus (MD) contains medial and lateral subregions (MDM/MDL), which are reciprocally connected to the vmPFC and dmPFC, respectively. These circuits maintain activity during delay periods in cognitive control tasks and are thought to guide correct behavioral output by maintaining a representation of task rule[13–17]. Likewise, the dorsomedial and ventromedial striatum (DMS/VMS) have both been linked to inhibitory control and attention[18,19], and receive input from the dmPFC and vmPFC, respectively. Moreover, specific mPFC→DMS projections are linked to the development of cognitive control and increased delay activity[12,20], whereas mPFC→VMS projections are associated with anticipation and reward processing during cognitive control tasks[11,21,22]. This indicates that prefrontal populations can be separated based on projection targets and that they are distinctly involved in behavior. However, the exact role and timing of activity of these pathways in cognitive control are unknown.

We provide evidence for the existence of four distinct prefrontal efferent pathways, which are involved in inhibitory control and attention. Neuroanatomical tracing using a retrograde virus and retro beads was used to identify corticothalamic and corticostriatal projection neurons. We then report distinct postsynaptic responses to prefrontal stimulation in striatal and thalamic neurons. Next, we measured behavioral performance in rats using a self-paced 5-choice serial reaction time task (CombiCage;[23]). We then tested the causal role of each projection in attention and inhibitory control using chemogenetics, which suggested distinct roles in inhibitory control depending on the projection target and population location in the mPFC. Finally, we investigated temporal dynamics of brain activity during task performance and found that the projection population had distinct activation patterns. Collectively, we here demonstrate a distinct role for each projection pathway in cognitive control.

## Results

**Distinct distribution of prefrontal projection neurons.** Pyramidal neurons projecting to the MD and striatum are located across the dmPFC and vmPFC[8,9]. However, whether these neurons belong to distinct projection populations is unclear. Therefore, we first expressed eYFP in the dmPFC or vmPFC and observed axonal eYFP expression in MD and striatum subdomains (Fig. S1a–d). We next infused retrobeads in the MD and striatum subdomains with a high degree of eYFP-positive axons. Quantification of labeled mPFC somata across three anterior-posterior-locations revealed a gradient of retrobead-positive neurons along the dorsoventral axis, as well as a gradient across cortical layers. We found that $90 \pm 2.25\%$ ($490.77 \pm 39.35$ cells/mm$^2$) of MDL-projecting neurons were in dmPFC areas, with the remaining cells ($52.93 \pm 13.38$) located in the vmPFC (Fig. 1a), and $81 \pm 3.07\%$ ($311.31 \pm 33.19$) of all MDM-projecting neurons were found in the vmPFC with the remaining neurons ($76.64 \pm 21.28$) situated in the dmPFC (Fig. 1b). MD-projecting mPFC neurons were primarily found in deep layers, while striatum-projecting mPFC neurons were located in layers 2/3 and 5. Of all DMS-projecting neurons, $82 \pm 1.6\%$ ($707,17 \pm 25.98$) were located in the dmPFC with the remaining part ($148.87 \pm 16.35$) being in the vmPFC (Fig. 1c). Of all VMS-projection neurons, $75 \pm 1.98\%$ ($380.42 \pm 58.30$) were located in the vmPFC with the remaining neurons ($132.20 \pm 13.71$) located in the dmPFC (Fig. 1d). Layer distributions of MDL-projecting and DMS-projecting neurons in the dmPFC, and MDM-projecting and VMS-projecting neurons in the vmPFC were significantly different (MDL/DMS: $\chi^2[2] = 54.97$, $p < 0.0001$; MDM/VMS: $\chi^2[2] = 103.80$, $p < 0.0001$). Neuron distribution revealed by retrobead labeling was confirmed by injection of retrograde CAV2-Cre in target areas combined with cre-dependent eYFP expression in the mPFC (Fig. S1e–h).

Cortical neurons can project to multiple target regions through axon collaterals[24,25]. Moreover, while projection neuron location was biased to layers, the layers did not exclusively contain neurons projecting to a single target area (Fig. 1a–d). To test whether single neurons project to both the MD and striatum, we separately injected CAV2-cre and retro-FLPo in the MD and striatum combined with cre-dependent eYFP expression and FLPo-dependent mCherry expression in the mPFC (Fig. 1e–f). Only a minority of dmPFC neurons (0.80%, 6/747) and vmPFC neurons (0.64%, 7/1101) were positive for both mCherry and eYFP (Fig. 1e–f), suggesting that most neurons specifically project to either the MD or striatum. In addition, no eYFP-positive or mCherry-positive neurons were positive for GAD-67, excluding the possibility that long-range interneurons were present[26]. Together, these data suggest that the majority of MD-projecting and striatum-projecting mPFC neurons form largely distinct pyramidal neuron populations.

**Distinct functional properties of mPFC output pathways.** Next, we wanted to investigate if the difference between mPFC output pathways to the MD and striatum were also reflected in the postsynaptic neuronal properties. If and how these mPFC-innervated neurons differ across the subdomains of the MD and striatum is poorly understood. Therefore, we tested whether postsynaptic dmPFC → MDL, vmPFC → MDM, dmPFC → DMS, or vmPFC → VMS neurons showed differential synaptic input properties, passive and active electrophysiological properties that could contribute to differential information processing in support of behavior. We performed whole-cell recordings in acute thalamic or striatal slices from animals injected with AAV9-Syn-Chronos-GFP in dmPFC or vmPFC (Fig. 2a, c). The mPFC and MD are interconnected through dense reciprocal connections[24]. Therefore, we additionally injected red retrobeads in the mPFC to target reciprocally connected MD neurons and recorded light-evoked postsynaptic currents. To prevent overstimulation, we adjusted the light intensity of the first pulse to an intensity that approximated a half maximum peak amplitude in the evoked postsynaptic current. Excitatory inputs from both dmPFC and vmPFC neurons to MDL and MDM neurons showed pronounced paired-pulse facilitation (Fig. 2b). In the striatum, we targeted medium spiny neurons and recorded light-evoked postsynaptic currents. Excitatory inputs from vmPFC to VMS showed paired-pulse depression, while

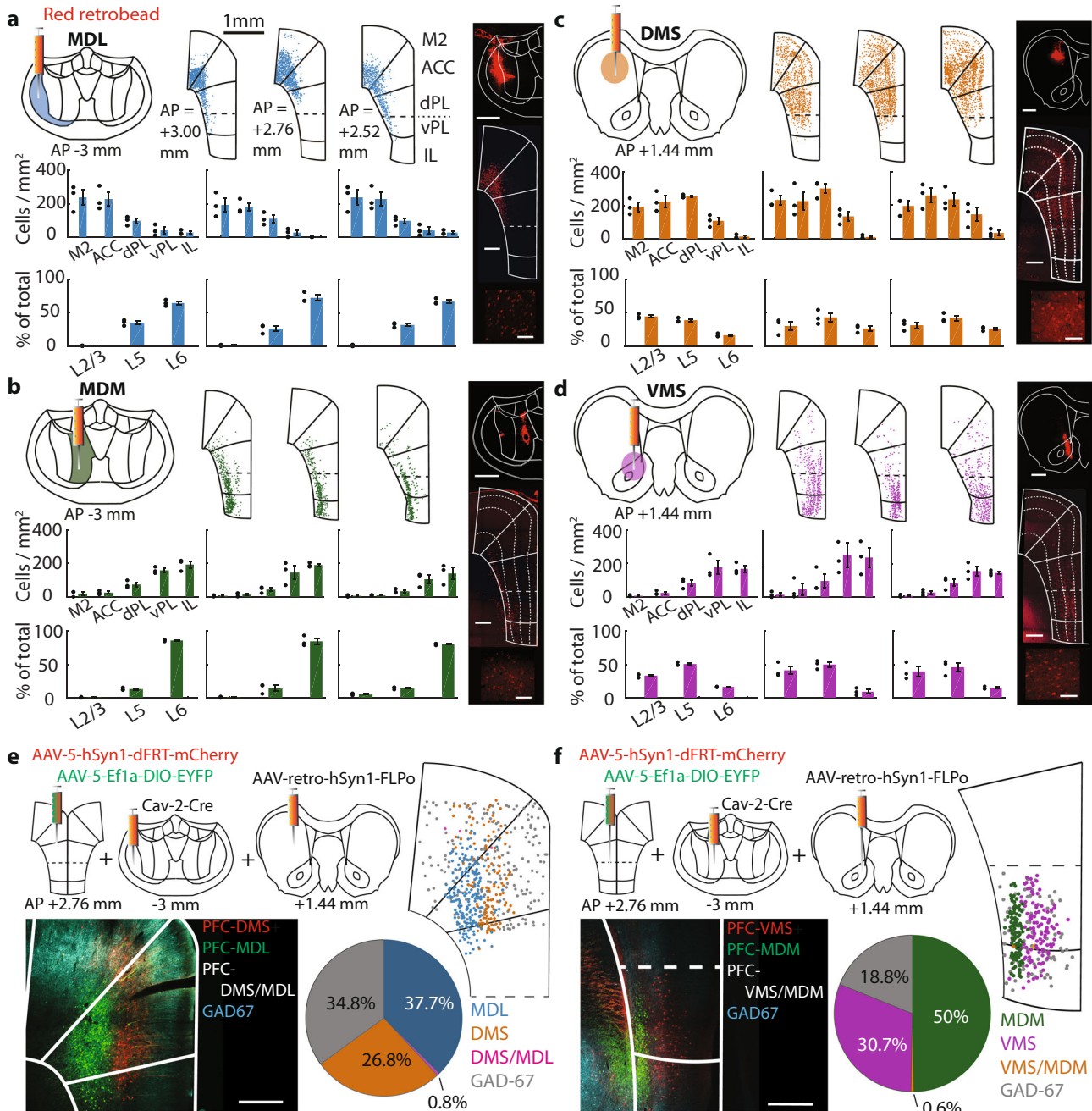

**Fig. 1 Distinct distribution of prefrontal projection neurons. a** Distribution of MDL-projecting neurons at three mPFC (anterior-posterior (AP) locations relative to bregma. Top left: retrobead injection location. Top middle: retrobead-labeled somata (representative example). Middle row: Labeled neuron distribution across mPFC. Bottom row: Neuron distribution across layers. Top right: representative example of retrobead labeling in MDL. Scale 500 μm. Middle right: representative example of labeled mPFC cells. Scale 500 μm. Bottom right: close-up of mPFC cell bodies. Scale 200 μm. Bar graphs represent mean ± SEM. **b–d** Similar to **a**, but for MDM-projecting neurons (**b**), DMS-projecting neurons (**c**), and VMS-projecting neurons (**d**). Data points in graphs represent individual rats (n = 3 rats). **e** Distribution of MD-projecting and striatum-projecting mPFC neurons in dmPFC. Top left: virus injection protocol. Bottom left: eYFP, mCherry, and GAD-67 stainings. Top right: labeled neuron distribution. Pie chart: quantification of projection neurons (blue: MDL-projecting neurons, pink: DMS + MDL-projections, orange: DMS-projections, gray: GAD-67-positive). Scale 500 μm. **f** Same as **e**, but for vmPFC neurons projecting to VMS and MDM. Pie chart: projection neuron quantification (green: MDM-projecting neurons, orange: MDM + VMS-projections, purple: VMS-projections, gray: GAD-67-positive). Scale 500 μm. Data in **e** and **f** each represents stainings in 2 rats.

the overall mean of dmPFC to DMS synaptic inputs showed no facilitation (Fig. 2d).

Next, we compared passive and active electrophysiological properties of the same postsynaptic neurons in the MD and striatum. Input resistance, membrane time constant (tau), capacitance, and sag ratio were determined using hyperpolarizing steps from −70 mV in the current-clamp configuration. MDM

neurons that were reciprocally connected to the vmPFC showed larger input resistance and larger membrane time constant compared to MDL neurons, while capacitance and sag ratio were similar (Fig. 2e–f, Fig. S2a–b). DMS and VMS neurons showed no differences in passive and active electrophysiological properties (Fig. 2g–h, Fig. S2c–d). The input-output relationship was tested using depolarizing steps from −70 mV in the current-clamp

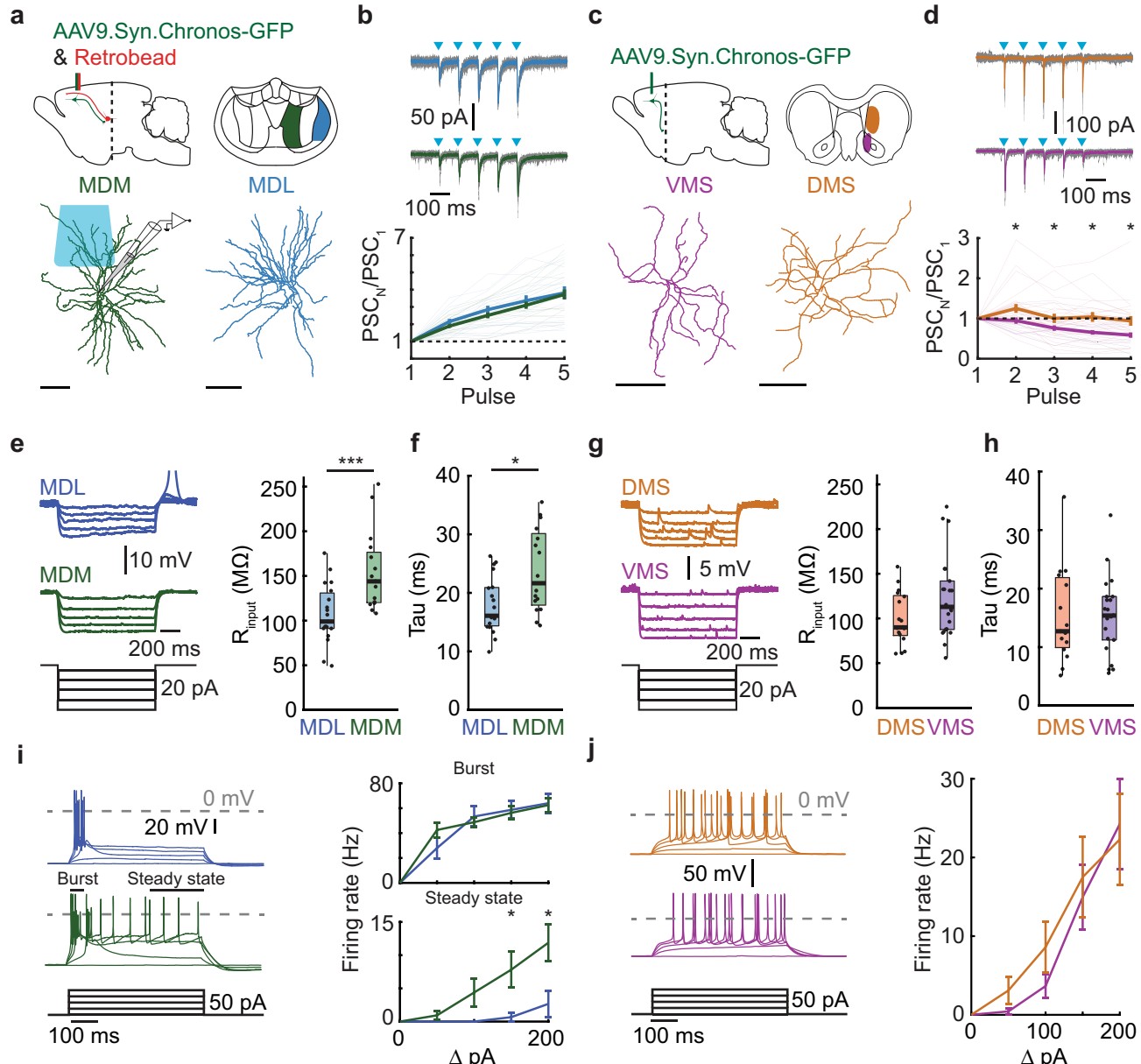

**Fig. 2 Distinct functional properties of mPFC output pathways. a** Synaptic input from mPFC to reciprocally-connected thalamic relay neurons and striatal medium spiny neurons. Bottom: Digital reconstruction of recorded neurons. Scale 100 μm. **b** Top: Example PSCs after Chronos activation (blue squares). Gray trace: individual sweeps; solid trace: median. Protocol: 5 pulses, 10 Hz, 1 ms. Bottom: Summary plot of paired-pulse ratios. MDL: $\chi^2[4] = 62.44$, $p < 0.0001$. MDM: $\chi^2[4] = 66.57$, $p < 0.0001$. Post hoc: $PSC_1$ vs. $PSC_{3-5}$, MDL: $p = 0.0002$, $p < 0.0001$, $p < 0.0001$, MDM: $p = 0.0003$, $p < 0.0001$, $p < 0.0001p$. **c** Same as **a**, for striatal neurons. **d** Same as **b**, for striatal neurons. DMS: $\chi^2[4] = 15.70$, $p = 0.0035$; VMS: $\chi^2[4] = 44.90$, $p < 0.0001$, Post hoc: VMS, $PSC_1$ vs. $PSC_{3-5}$, $p = 0.0206$, $p < 0.0001$, $p < 0.0001$; VMS vs. DMS, $stim_{2-5}$, $p = 0.0113$, $p = 0.0484$, $p = 0.0113$, $p = 0.0133$. **e** Input resistance ($R_{input}$). Left: example traces. Right: summary plot. $p_{MDL-MDM} = 0.0005$. **f** Membrane time constant (Tau). $p_{MDL-MDM} = 0.0120$. **g** Input resistance, striatal neurons. **h** Membrane time constant. striatal neurons. **i** Action potential firing profiles, 0–200 pA current steps. Left: example traces. Burst, 50 ms after the first spike, Steady-state, last 200 ms of the pulse. Right: Summary plot (burst). $MDL_{Burst}$: $\chi^2[4] = 36.95$, $p < 0.0001$, $MDM_{Burst}$: $\chi^2[4] = 47.32$, $p < 0.0001$; $MDM_{SteadyState}$: $\chi^2[4] = 30.01$, $p < 0.0001$; Post hoc, 150 pA: $p_{MDL-MDM} = 0.0192$, 200 pA: $p_{MDL-MDM} = 0.0192$. **j** Same as **i**, for striatal neurons. DMS: $\chi^2[4] = 44.39$, $p < 0.0001$, VMS: $\chi^2[4] = 69.54$, $p < 0.0001$. Data in **b**, **d**, **e**, **f**, **g**, **i** and **j** are mean ± SEM. Boxplots: center line, median; edges, 1st and 3rd quartile; whiskers, range without outliers. Mann–Whitney tests in **d**, **e**, **f**, and **h** were two-sided. Friedman with Dunn's was used in **b**, **d**, **i**, and **j**. Significance: $*p < 0.05$, $***p < 0.001$. Group sizes: MDM ($n = 18$ cells, 6 rats), MDL ($n = 19$ cells, 7 rats), VMS ($n = 21$ cells, 6 rats), DMS ($n = 23$ cells, 7 rats).

configuration. In correspondence with their higher input resistance, MDM neurons showed an increased steady-state action potential firing rate in response to depolarizing current steps, but no change in burst firing, compared to MDL neurons (Fig. 2i) Both dorsal and ventral striatal neurons showed a similar increase in depolarizing current-evoked firing rates (Fig. 2j).

Together this data shows that the four mPFC output pathways to the MD and striatum have differential postsynaptic input properties and electrophysiological properties. These different characteristics could support differential integration of mPFC neuronal activity within MD or striatal subregions in support of behavior.

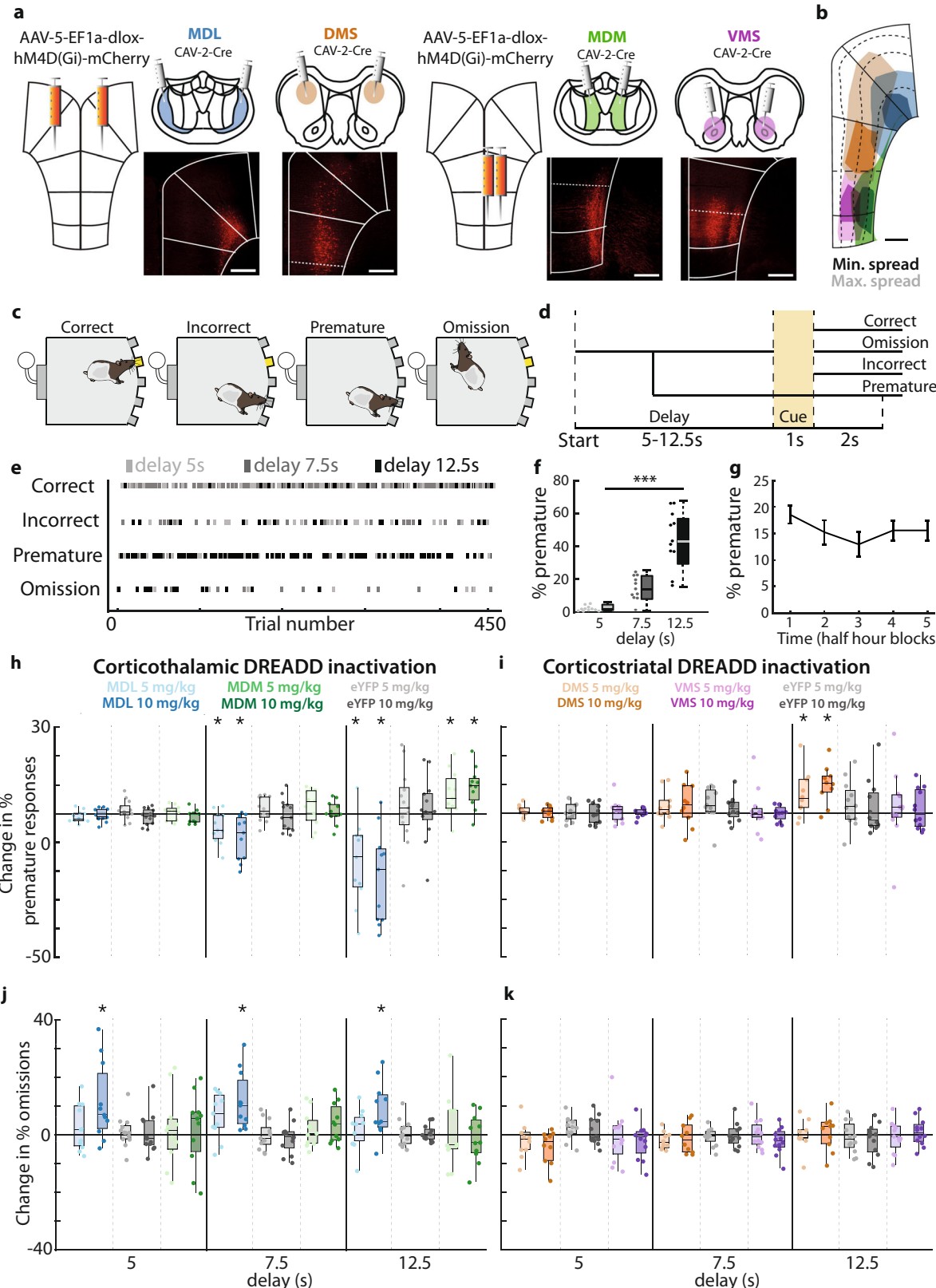

**Bi-directional regulation of inhibitory control by mPFC projection neurons.** To test whether these four mPFC output pathways to the MD and striatum indeed supported differential roles in inhibitory control and attention, we selectively silenced these pathways during behavior. The mPFC, MD, and striatum regulate cognitive control of behavior[4,16–18], but the role of

specific mPFC projections to MD and striatal subdomains is incompletely understood. We expressed the inhibitory DREADD-receptor hM4D(Gi) in each projection population to test whether they are causally involved in cognitive control (Fig. 3a). Clozapine-N-Oxide (CNO) elicited membrane potential hyperpolarization, increased rheobase, and decreased spike frequency

**Fig. 3 Bi-directional regulation of inhibitory control by mPFC projection neurons. a** Viral injection protocol for hM4D(Gi)-expression in projection neurons. Bottom: Representative example of mPFC hM4D(Gi)-expression. **b** Spread of hM4D(Gi)-expression. **c** Possible 5-CSRTT trial outcomes. **d** Schematic representation of possible 5-CSRTT outcomes. Delay was randomly varied between 5, 7.5, and 12.5 s in variable delay sessions. **e** Example of behavioral responses during a single variable delay session. Dots indicate individual trials, darker colors represent longer delay trials. **f** Premature responses in variable delay sessions. $F[2,20] = 51.13$, $p < 0.0001$. **g** Distribution of premature responses after saline and CNO injections in 2.5-h variable delay session, divided into 30-min blocks. **h** Change in premature responding of animals expressing hM4D(Gi) or eYFP in MDL-projecting or MDM-projecting neurons in variable delay sessions. MDL: $F[8,128] = 9.31$, $p < 0.0001$, $p_{CNO5} = 0.0049$, $d_{CNO5} = 0.99$, $p_{CNO10} = 0.003$, $d_{CNO10} = 1.36$. MDM: $F[8,128] = 9.31$, $p < 0.0001$, $p_{CNO5} = 0.021$, $d_{CNO5} = 0.69$, $p_{CNO10} = 0.0044$, $d_{CNO10} = 1.00$. **i** Change in omissions in MDL-projecting or MDM-projecting neurons. MDL: $F[4,64] = 4.25$, $p = 0.004$. **j** same as **g**, for animals expressing eYFP and hM4D(Gi) receptors in DMS-projecting and VMS-projecting neurons. DMS: $F[8,124] = 2.72$, $p < 0.0001$, $p_{CNO5} = 0.047$, $d_{CNO5} = 0.51$, $p_{CNO10} = 0.0039$, $d_{CNO10} = 0.80$. **k** same as **h**, for DMS-projecting and VMS-projecting neurons. Dots represent individual animals; bar graphs represent mean ± SEM. Scale in **a** and **b**: 500 µm. Boxplots in **f**, **h–k**: center line, median; box edges, 1st and 3rd quartile; whiskers, data range without outliers. One-way ANOVA was used in **f**. Three-way mixed repeated-measures ANOVA was used in **h**, **i**, **j**, and **k**, with FDR-corrected, paired one-way post hoc tests $t$-test vs. saline. Significance: $*p < 0.05$, $***p < 0.001$. Group sizes: MDL ($n = 11$ rats), MDM ($n = 11$), eYFP-Thalamus ($n = 13$), DMS ($n = 10$), VMS ($n = 12$), eYFP-Striatum ($n = 12$).

under current step injections in acute mPFC brain slices of hM4D (Gi)-expressing animals (Fig. S3).

DREADD-expressing animals were trained in the CombiCage 5-CSRTT[23]. In this modified, self-paced and semi-automatic version of the 5-CSRTT, the homecage of the animal was linked to the operant cage. This allowed rats to progress through the task at their own pace, resulting in a large number of daily trials, and little need for human interference. Animals could earn food rewards by responding to a visual cue that appeared randomly in one of five cue holes (Fig. 3c). Premature responses made before cue onset were used as a measure for inhibitory control, whereas attention was measured using the ratio of correct and incorrect responses or as a percentage of trials with omitted responses (Fig. 3c, d). After reaching stable baseline task performance, animals started testing sessions. On test days, we varied delays between the trial start and cue presentation. This increases cognitive load and avoids overtraining, and allows more specific investigation of attention and inhibitory control, respectively[23,27]. Animals were injected with each CNO dose in a randomized order and performed $402 \pm 10$ trials (mean ± SEM) per 2.5-h session in these conditions (Fig. 3e). Premature responding consistently increased with longer delay duration (Fig. 3e, f, Table S1–6), while trials with shorter cue duration decreased accuracy and increased omissions (Fig. S4a, Table S8–13).

CNO-mediated inhibition of MDL-projecting mPFC neurons decreased premature responding, especially in trials with long delays (Fig. 3h; values relative to saline condition.). In addition, we observed a delay-independent increase in omissions (Fig. 3j, Table S1). To test whether CNO effects persisted the entire 2.5 h session, we analyzed sessions in five 30-min blocks. The decrease in premature responses and increase in omissions were consistent (Fig. S4c–d), indicating that CNO effects lasted throughout the entire session. In contrast to inhibition of MDL-projecting neurons, perturbation of vmPFC→MDM projections increased premature responding, but without change to omissions (Fig. 3h, j, Table S2). In a different set of sessions, we varied the cue duration and not the delay. CNO-mediated inhibition of MDL-projecting neurons during these sessions increased omissions, independent of cue duration, but did not affect the accuracy or any other measured behavioral parameter. No effect was found for MDM-projecting neurons (Fig. S4m, o, Tables S8–9). To exclude confounding effects of motivation and motor control we tested for effects of CNO on 5-CSRTT parameters such as response latency or the number of started trials. These were unaffected in all sessions (Table S7). Finally, no effect of CNO was observed in the eYFP control group (Fig. 3h, j, Table S3, S7, S10), excluding the possibility of non-specific effects of CNO. Together, these data show that mPFC projections to thalamic subdomains have opposite roles in cognitive control: inhibition of vmPFC→

MDM projections increases premature responses, whereas inhibition of dmPFC→MDL projecting populations reduces premature responses.

Inhibition of DMS-projecting dmPFC neurons increased premature responding, but did not affect omissions (Fig. 3i, k, Table S4), while inhibition of vmPFC→VMS projections did not affect premature responses, omissions, or any other behavioral parameter in the task (Fig. 3i, k, Table S5, S12, S14). During variable cue duration sessions, CNO had no effect on accuracy (Tables S11–13), and additional behavioral parameters such as premature responses, response latencies, and the number of started trials were also unaffected (Tables S11–14), suggesting that inhibitory control, motivation, and task engagement of animals were unaltered. Altered premature responding can reflect changes in temporal strategies or perception[28]. However, we found no effect of CNO on the temporal distribution of premature response latencies in long delay trials in variable delay sessions (Fig. S4g–l), suggesting that the temporal structure of responding was unaffected.

Thus, while dmPFC projection neurons to the MD and striatum bi-directionally guide inhibitory control, in the vmPFC only projection neurons that target the MD are involved. Thereby, the dmPFC and vmPFC can orchestrate response inhibition in opposite manners controlling distinct thalamic subregions. In addition, dmPFC neurons in different cortical layers can achieve this, through opposite control of thalamic and striatal regions.

**mPFC projections show distinct activation during inhibitory control.** Previously, we showed that distinct prefrontal projection populations can guide inhibitory control. Next, we investigated population activity during the delay periods of 5-CSRTT trials. Prefrontal neurons show various activity patterns during 5-CSRTT trials with distinct behavioral outcomes[3,6,19,21]. To determine the activation profiles of the projection populations targeting the MD and striatum during 5-CSRTT trials, we expressed GCaMP6m in each population (Fig. 4a–c). Using fiber photometry, we recorded GCaMP6m fluorescence across several variable delay sessions, during which animals started up to $201 \pm 5$ trials per 1-h session (Fig. 4d–e, Fig. S5). Animals increased premature responding in long delay duration trials (Fig. 4f, S5). Fluorescence changes reflecting neuronal activation recorded during behavioral trials closely followed delay period duration, with fluorescence signal elevation lasting longer during longer delay trials (Fig. 4e–g). The area under the curve (AUC) of fluorescence between the trial start and cue presentation was significantly larger in long delay-trials in all populations, indicating that increased activity was strictly related to periods of attention and inhibitory control (Fig. 4g–h). Both dmPFC→ MDL and dmPFC→DMS projections showed stronger

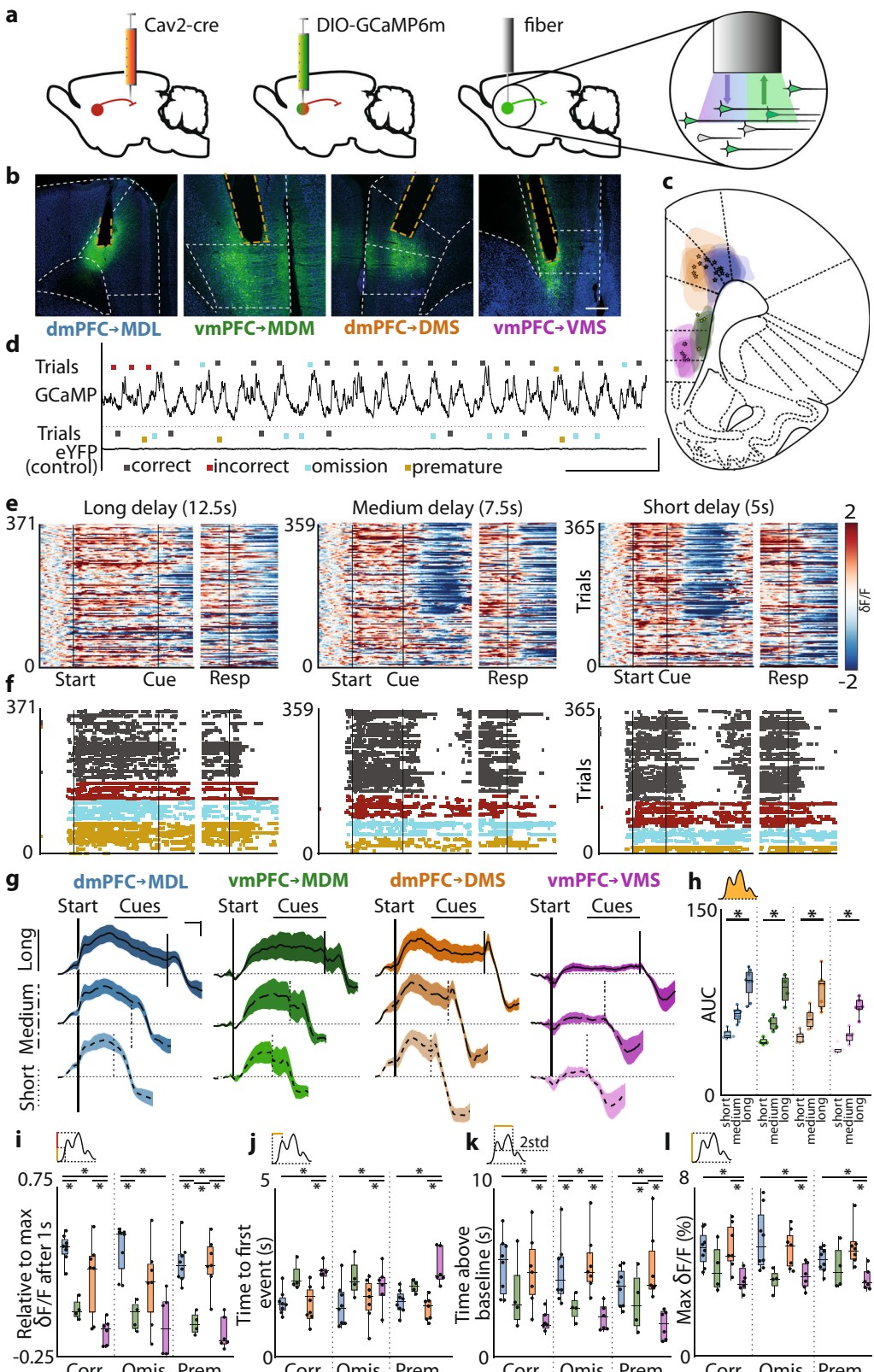

activation in the first second following trial start than vmPFC projection populations (Fig. 4i,). Ventral mPFC → VMS neurons showed significantly less activation compared to other projection neuron populations (Fig. 4i–l). Populations of dmPFC neurons targeting the MDL and DMS reached peak fluorescence, defined as the first local maximum above 20% of overall peak

fluorescence, faster than vmPFC projection neurons targeting the VMS (Fig. 4j). Dorsal mPFC projection neuron populations targeting MDL and DMS were also activated longer and for a greater proportion of the delay period (Fig. 4k), and reached higher relative fluorescence values than vmPFC neuron populations (Fig. 4l).

**Fig. 4 mPFC projections show distinct activation during inhibitory control. a** Injection and fiber placement protocol. **b** Representative example of GCaMP6m-expression and fiber placement. **c** Expression and fiber placement at AP + 2.76. Shaded areas: GCaMP6-expression. Stars: fiber tips. **d** Example traces from GCaMP6m-expressing and eYFP-expressing animals. Dots represent trialstarts. **e** Variable delay trials from one rat, by the trial outcome. Left two plots: 5 s delay trials, synchronized on trialstart (left) or response (right). Middle: 7.5 s delay. Right: 12.5 s delay. δF/F z-scored to trial baseline (5 s–1 s before trialstart). **f** Same as **e**, color-coded by the outcome. Dots represent frames with δF/F > 2std above baseline. Colors same as in **d**. **g** Average correct trial fluorescence. **h** Area under the curve (AUC) during the correct trial delay. Delay, short vs. long, $p_{MDL} = 0.0002$; $p_{MDM} = 0.014$; $p_{DMS} = 0.0006$; $p_{VMS} = 0.0016$. **i** Fluorescence rise kinetics 1 s after trialstart, a fraction of peak fluorescence. Correct, $p_{MDL-MDM} < 0.0001$; $p_{DMS-VMS} < 0.0001$; $p_{MDL-VMS} = 0.0095$; Omissions, $p_{MDL-MDM} = 0.002$, $p_{MDL-VMS} = 0.0029$; Premature, $p_{MDL-MDM} = 0.0006$, $p_{MDL-VMS} = 0.0001$, $p_{DMS-MDM} = 0.0017$, $p_{DMS-VMS} = 0.0002$. **j** Time to the first synchronous event. Correct, $p_{MDL-VMS} = 0.0007$, $p_{DMS-VMS} = 0.006$; Omissions, $p_{MDL-VMS} = 0.0344$, $p_{DMS-VMS} = 0.0344$; Premature $p_{MDL-VMS} = 0.0102$, $p_{DMS-VMS} = 0.002$. **k** Total time signal > baseline + 2std. Correct $p_{MDL-VMS} = 0.0093$, $p_{DMS-VMS} = 0.0241$; Omissions $p_{MDL-MDM} = 0.0285$, $p_{MDL-VMS} = 0.0127$, $p_{DMS-VMS} = 0.0285$; Premature $p_{MDL-VMS} = 0.0008$, $p_{MDL-VMS} = 0.0383$, $p_{DMS-VMS} = 0.0008$. **l** Peak fluorescence during the delay. Correct, $p_{MDL-VMS} = 0.0215$, $p_{DMS-VMS} = 0.0493$; Omissions, $p_{MDL-MDM} = 0.0448$, $p_{MDL-VMS} = 0.034$, $p_{DMS-VMS} = 0.034$; Premature, $p_{MDL-VMS} = 0.0294$, $p_{DMS-VMS} = 0.0078$. Dots represent individual animals in **h–l**. Shades in **g** represent SEM. Scale in **e** is the same for all heatplots. Boxplots in **h–l**: center line, median; edges, 1st and 3rd quartile; whiskers, range. Friedman with Dunn's used in **h**. Kruskal–Wallis with uncorrected Dunn's and FDR used in **i–l**. Significance: *$p < 0.05$. Scales: (**a**) 400 µm, (**d**) horizontal 50 s, vertical 10% δF/F, (**g**): horizontal 2 s, vertical 1% δF/F. Group sizes: MDL ($n = 8$ rats), MDM ($n = 4$), DMS ($n = 7$), VMS ($n = 6$).

Each mPFC projection pathway showed delay-dependent activity. More specifically, dmPFC projections appear to be activated more rapidly after trial initiation than vmPFC projections. In addition, dorsal projection pathways remain active for a longer portion of the delay period. These effects were consistent for each type of trial outcome. These results show that dorsal and ventral mPFC projection neuron populations to MD and striatum show distinct activation profiles during attention and inhibitory control.

**The activity of mPFC projection neurons encodes behavioral trial outcome.** Prefrontal projection populations are active during the 5-CSRTT. The amplitude of neuronal activity rates during the delay period has been linked to trial outcome[3,29]. To test whether activity patterns of specific mPFC projection neuron populations encode trial outcome, z-scored mean fluorescence traces were compared to a randomly resampled population using a bootstrap approach to determine periods of elevated activation during long-delay trials (Fig. 5a–c, upper panel. See "Methods" section for more information about this test and parameters used). Populations of MDL, MDM, and DMS projection neurons each showed a specific temporal activation window, in particular early in the delay period, and before a response. Ventral mPFC → VMS projection neurons showed no elevated fluorescence. We then tested whether the activation patterns differed between projection populations using permutation tests (Fig. 5c, lower panel. Test parameters: iterations = 5000, significance threshold $\alpha = 0.01$. See "Methods" section for more information about this test and parameters used). Projection populations significantly differed in activity during several temporal windows, in particular, compared to VMS-projecting neurons (Fig. 5c). Mediodorsal thalamus-projecting populations (dmPFC → MDL and vmPFC → MDM) showed differences in activation during the delay period in correct, omission and premature response trials, whereas dmPFC populations projecting to MD or striatum did not show differences in activation (Fig. 5c). Ventral mPFC populations projecting to VMS and MDM neurons showed distinct activation during the delay period in omission and premature response trials (Fig. 5c). Finally, in no population did we find significant activity leading up to the cue in omitted trials. These data show that MDL-projection, MDM-projection, and DMS-projection neurons were activated during cognitive control of behavior with projection-specific activity dynamics.

We then asked whether these projection population-specific activation profiles encode behavioral trial outcomes. We compared activation dynamics within each population during trials with different behavioral outcomes (Fig. 5d) and bootstrapped

differences between activity windows (Fig. 5e, upper panel). Projection populations showed distinct windows of elevated activation during correct, omission, and premature response trials during the delay period and around task-relevant events. Statistical comparison using permutation tests (Fig. 5e, lower panel) showed that dmPFC → MDL neurons were more active during the delay in correct trials compared to premature responses, indicating that this population is associated with trial outcome. In contrast, ventral mPFC neurons projecting to the MDM showed reduced activation both during omission trials and premature response trials compared to correct trials (Fig. 5e). Dorsal mPFC neurons projecting to the striatum showed brief predictive windows during the delay period when comparing correct response and omission trials (Fig. 5e). Together, these results show that MDL-projection, MDM-projection, and DMS-projection neurons are involved in attention and inhibitory control and they each contain predictive information to predict the trial outcome.

## Discussion
In this study, we provide anatomical, in vitro electrophysiological, behavioral, and neurophysiological evidence for distinct roles of four distinct prefrontal projection pathways in behavior (Fig. 6). Projection neuron populations are spatially segregated in the mPFC, and inhibition of these projection neurons disrupts both inhibitory control and attention. We show for the first time that mPFC projection neurons targeting distinct MD subregions have opposite roles in inhibitory control. We also show that thalamus-expressing and striatum-projecting mPFC neurons have distinct roles in inhibitory control. Moreover, projection neuron populations showed distinct temporal dynamics that predicted behavioral trial outcomes. Finally, we show that postsynaptic neurons in target regions respond to prefrontal input in a distinct manner. Taken together, we present four lines of experimental evidence for distinct roles for mPFC projection neuron populations in cognitive control.

Prefrontal neurons are known to project to the striatum or thalamus in a dorsal-to-ventral and layer-based distribution[8,9,11,12,22,30]. Our results corroborate earlier findings in corticostriatal anatomy[9], where VMS-projecting and DMS-projecting neurons are abundant in layers 2, 3, and 5, and scarce in layer 6. However, compared to that study, we do report a larger proportion of DMS-projecting neurons in layers 2 and 3, which could potentially be explained by the relatively limited projection area we targeted in the DMS. It has been shown that dmPFC neurons project to various striatal regions[31], hence it may well be that neurons that project to the most dorsomedial regions of the

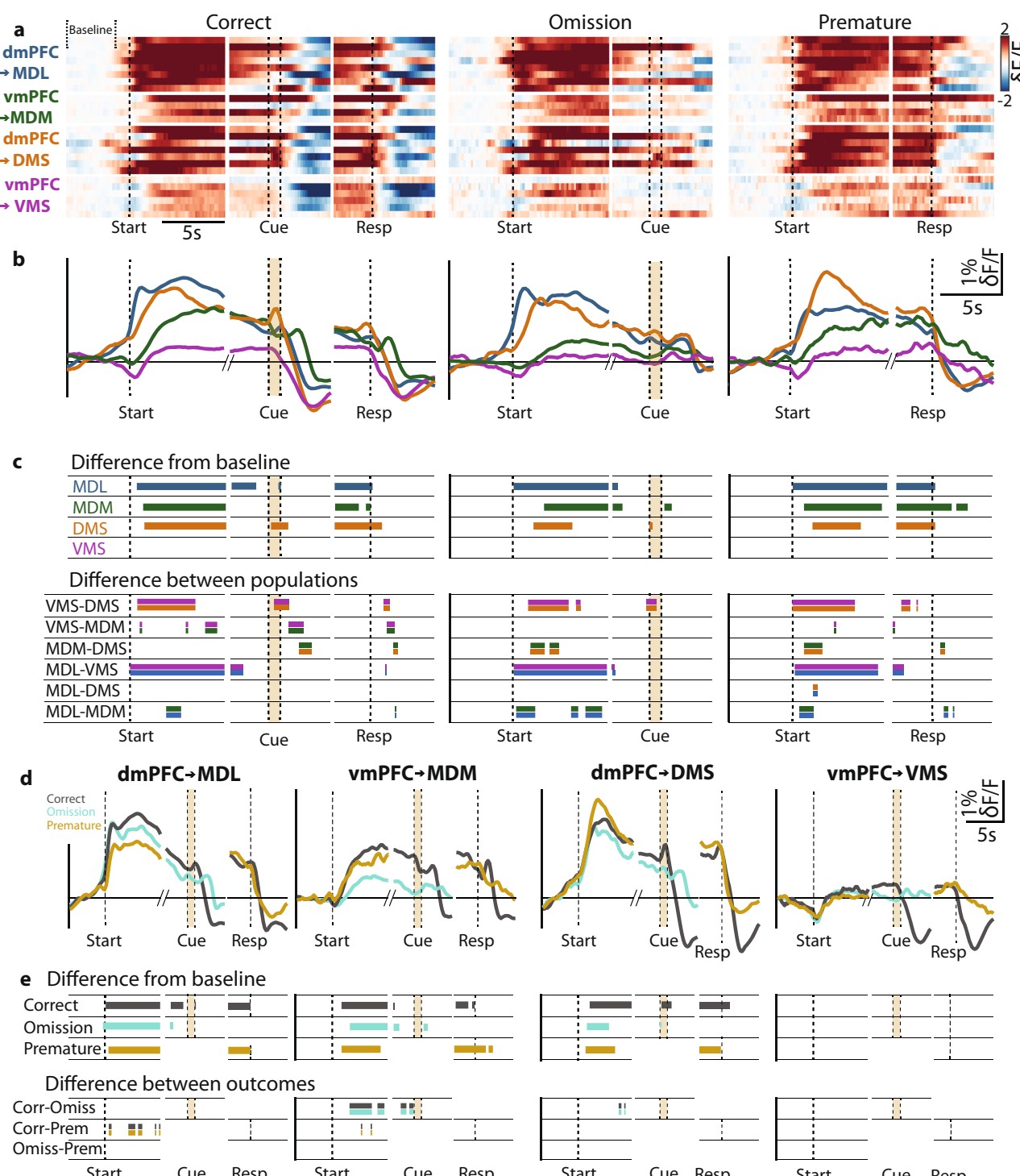

**Fig. 5 Activity of mPFC projection neurons encodes behavioral trial outcome. a** Average GCaMP6 fluorescence of each mPFC projection neuron population in individual rats during the correct response, omission, and premature response trials (δF/F is z-scored to trial baseline). The plot is capped at −2 and 2 z-scores. Baseline marked in the top left. **b** Group activity during behavioral trials. **c** Upper: windows of significantly increased activity during the delay and around cue and response. Bootstrap parameters: 5000 iterations, $\alpha = 0.001$. Lower: windows with a significant difference in activity between indicated projection populations. Bars represent significant permutation test results. Double-colored bars represent populations that were compared. **d** Average activity during different trial outcomes. **e** Statistical evaluation of activity in **e**. Upper: time windows with significantly elevated activity during the delay, and around cue and response. Bootstrap parameters same as in **c**, see Supplemental Methods section for the detailed procedure. Lower: windows with a significant difference between activities during different behavioral trial outcomes. Bars represent significant permutation test results. Permutation test parameters same as in **c**. Singleton significant data frames were discarded. Double colored bars as in **c**. The color scale in the top left of **a** is the same for all heat plots. Group sizes and colors: MDL (blue, $n = 8$), MDM (green, $n = 4$), DMS (purple, $n = 7$), VMS (orange, $n = 6$).

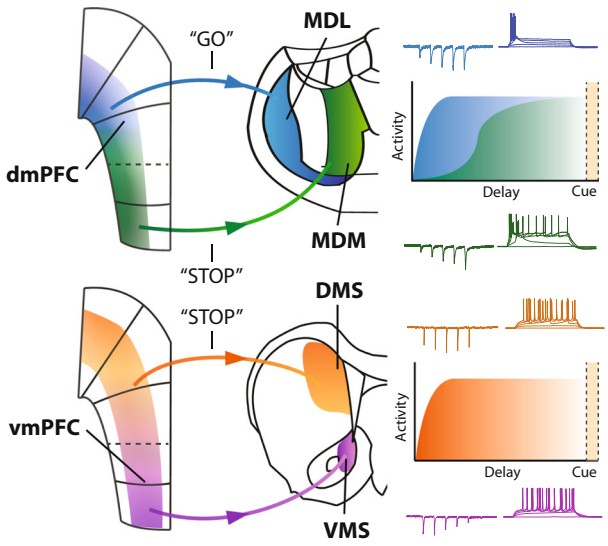

**Fig. 6 Corticostriatal and corticothalamic pathways in cognitive control.** Schematic overview of the mPFC output pathways. The dmPFC → MDL pathway encodes a behavioral "go"-signal, and projections from the vmPFC → MDM encode a "stop"-signal. Relay neurons in the MD show a facilitating response to prefrontal input, and the dmPFC → MDL pathway has quicker kinetics than vmPFC → MDM projections. Projections from dmPFC to DMS represent a "stop"-signal, while we found no behavioral effects or significant activation associated with the vmPFC → VMS pathway. Postsynaptic responses in striatal MSNs are depressing in response to vmPFC → VMS input, and we found no change in postsynaptic dmPFC → DMS responses. Colors, blue: dmPFC → MDL pathway, green: vmPFC → MDM pathway, orange: dmPFC → DMS pathway, purple: vmPFC → VMS pathway.

striatum are located more superficially. While it has previously been reported that mPFC projection neurons to the MDM and MDL are located mainly in deep layers[9], we here show that these projections are localized in distinct dorsoventral locations in the mPFC. Our data further specifies prefrontal afferents into populations of excitatory neurons that preferentially target subdomains of the thalamus and striatum. In addition, although inhibitory corticostriatal projections have been reported in mice[26], we found no GAD-67 expression in projection populations. Possibly, inhibitory projections target more posterior regions of the caudate-putamen[32]. While projection neurons were located mostly in specific prefrontal layers and subregions, we also show that they can be situated outside the regions we described. Projection-specific transcriptomic analysis of mPFC neurons[10,22,30] may resolve this issue. Furthermore, while we find little evidence for axon collaterals to both the MD and striatum, projection populations could be interconnected within the mPFC. This could potentially result in projection-unspecific effects on behavior and could be resolved by doing manipulations or recordings at axon terminals. Finally, we used both retro-AAV and CAV vectors, which can have distinct viral tropisms[33–35]. While, to our knowledge, no such effects have been reported for the populations we investigated, a method to circumvent this is to use an enhanced CAV vector[34].

Chemogenetic inhibition of mPFC projection neurons during variable delay 5-CSRTT sessions caused bi-directional effects on cognitive control. Premature responding decreased after inhibition of MDL-projecting neurons, but increased after perturbation of DMS-projection and MDM-projection activity. Only inhibition of MDL projections increased omissions. In addition, we observe no effects of CNO in the variable cue duration sessions. Animals

only made a small number of premature responses in this protocol, which did not include conditions (long delay) where we found CNO-mediated effects on premature responses. Studies that globally perturbed mPFC function through lesions, chemogenetics, or optogenetics during 5-CSRTT generally affect parameters reflecting attention and inhibitory control[1,2,4]. We perturbed the physiological activity of subpopulations within these larger mPFC regions. Our results suggest that inhibiting small and specific populations of projection neurons disentangles specific aspects of cognitive control such as inhibitory control and attention, which are collectively affected when manipulating entire mPFC subregions in a non-specific manner. Silencing multiple projection populations using multiple-wavelength optogenetics could provide further insight into the role of each projection and the redundancy of information sent through projections.

Activation of mPFC neurons during the delay period of cue detection paradigms has frequently been reported, but timing and amplitude of activity varies between trial outcomes, target area, and task parameters[3,5,20,21,36]. In all projection populations, we observed that activity followed delay duration, indicating that each population was activated in support of cognitive control over behavior. We find that dorsal mPFC projection neurons were recruited faster than ventral mPFC projection neurons, and those dmPFC projection neurons were active for a larger proportion of the delay period. This is in line with previous findings of different activity kinetics between non-identified vmPFC and dmPFC units and suggests a more proactive role of the dorsal mPFC and a more reactive role of the ventral mPFC[4,5]. We also report differences in population activity between projection populations, and between activity levels during delay periods leading up to different trial outcomes. Hence, fiber photometry recordings indicate that there is population activity that significantly deviates from baseline, and chemogenetic inhibition showed that disruption of such physiological levels of activity caused deficits in inhibitory control and attention. Optogenetic identification combined with optogenetic manipulation has been used to further characterize the role of projection populations in behavior[29] and would be a suitable technique to combine activity recordings with targeted inhibition.

Connections between the mPFC and MD are organized in recurrent loops, through which the MD can amplify local connectivity in the mPFC[13,24]. Both mPFC → MD projections and MD neurons have been associated with behavioral flexibility and working memory, and drive correct behavioral output in different paradigms by maintaining a representation of a task rule[13,14,37]. In addition, mPFC input to MDL neurons is required for proper rule encoding[38]. In this study, we show an opposite effect of manipulation of mPFC neurons projecting to MDL- or MDM when delay duration was unpredictable. In addition, inhibiting mPFC to MDL neurons increased omissions. Possibly, inhibition of this projection affects rule encoding, which thereby reduces response readiness and manifests behaviorally with both a reduction of premature responses and increased omissions. Hence, activity in the MDL-projecting population could drive responsive action, while MDM-projecting mPFC neurons could relay a signal to withhold a response until a sensory event occurs. Alternatively, mPFC → MD projections could be responsible for the maintenance of rule representation, rather than encoding. Perturbation of these projections specifically during rule encoding phases of the task could further unravel their exact role.

We find that prefrontal inputs elicit a facilitating response in both MDL and MDM neurons, which may be a potential mechanism through which recurrent activity in corticothalamic circuits is maintained during a delay period, and through which these projections regulate rule encoding or maintenance[38,39]. This

may also be a representation of prefrontal 'driver' and 'modulator' inputs to the MD. These inputs primarily originate in layer 5 and layer 6, respectively, and have been associated with distinct postsynaptic responses[40]. The anatomical positioning of MDM-projecting and MDL-projecting neurons resembles this distinction, which may indicate that the MDL receives more modulatory input, and MDM receives driving input. We also find that MDL-projecting neurons are recruited earlier during the delay than MDM-projecting neurons, supporting earlier evidence that MD subregions likely have distinct roles and are part of distinct circuits, and that dmPFC activity precedes vmPFC activity in cognitive control paradigms[4,5]. Our findings that MDL and MDM differ in basic electrophysiological features further support distinct roles in attention and inhibitory control. In addition, we show differences between population activity before premature and correct responses in the MDL, and between omissions and correct responses in the MDM. This difference suggests that different levels of activity in this circuit can underlie distinct types of behavior.

The mPFC has been shown to exert top-down control over the DMS[20], but chemogenetic and optogenetic inhibition of the dmPFC did not affect premature responding[2,4]. However, these manipulations were not targeted to a specific population and covered both MDL-projecting and DMS-projecting populations, which have an opposing effect. Our results show an increase in population activity in DMS-projecting neurons during the delay period. Changes in firing rate have been reported in both mPFC and DMS during the delay before a response[3,21,36,41], as well as during cue presentation[42]. Premature responses have been associated with reduced amplitude of neuronal activity in the dmPFC and in dmPFC → DMS projecting neurons[29,43]. Our data show that this population is active during the delay period and during the cue presentation before a correct response. While we did not find a reduced amplitude in the delay period before premature responses, we did see a shorter active window compared to correct responses. We also found a mixed synaptic input response in the DMS, which could be due to projection neurons differentially innervating D1-receptor and D2-receptor-expressing MSNs[44,45], or by specific topological innervation patterns seen in corticostriatal projection neurons[46]. Direct-pathway and indirect-pathway striatal medium spiny neurons have been hypothesized to represent competing "go" or "stop"-signals (see Cox and Witten, 2019). Top-down prefrontal input to the striatum is thought to guide the striatal bias into either of these behavioral outcomes[47,48]. Hence, a likely explanation is that the dmPFC→DMS input we observe guides the striatal network into a "stop" decision in the 5-CSRTT. Another potential mechanism could involve striatal dopamine. It was shown that optogenetic enhancement of mPFC excitability diminishes the striatal response to dopamine and suppresses reward-seeking behavior[49], while infusions of both D1 and D2-like receptor agonists specifically in the DMS increase premature responding in the 5-CSRTT[50]. Thereby, dopamine in the DMS may increase reward-seeking and impulsivity, which can be controlled by mPFC inputs in a top-down fashion. Previous work suggests that this projection may also be important for the accuracy of responding[51]. In our study, we targeted a more specific neuronal population, which could account for distinct behavioral effects.

We found no behavioral effect of inhibition of VMS-projection neurons. Previous functional disconnection studies targeting the mPFC and NAc shell, but not core, showed increased premature responding[52]. In addition, mPFC and contralateral NAc lateral core lesions increased premature responses after an error in the 5-CSRTT, suggesting a role of this pathway in adaptive control[53]. However, we did not target a specific NAc subregion. Our neuroanatomical data shows axon terminals in the medial ventral caudate-putamen and NAc. In addition, the NAc core receives top-down glutamatergic inputs from several other brain regions, such as the ventral hippocampus or insula[54,55]. It has been shown that fast-spiking interneurons in the NAc core have different levels of activity leading up to correct and premature responses in the 5-CSRTT, indicating that this area is active during the task[19]. We also find that NAc neurons show a depressing response to vmPFC input and that VMS-projecting neurons do show delay-dependent kinetics of population activity, even though signal amplitude was not significantly increased from baseline. Hence, the vmPFC does project to the NAc, but it likely does not drive the behavior we studied. While activity parameters at times do not significantly differ from other projections, it is likely that this activity is not synchronized enough to yield significantly elevated activity windows during the delay. Whether sparse mPFC → NAc activity is involved in cognitive control remains to be tested.

Together, our findings show a functional distinction between prefrontal projection populations, where MDL-projection neuron activity drives responses, and MDM-projections and DMS-projections withhold responses during a delay period. Populations have distinct patterns of activity during a 5-CSRTT trial and elicit distinct responses in postsynaptic neurons in the target area. This gives rise to a view of the prefrontal projection populations being central in several, but distinct pathways that lead to behavioral action or inhibition. This becomes especially relevant because abnormal prefrontal delta and theta activity have been associated with cognitive deficits in schizophrenia and Parkinson's Disease[56,57]. While we did not investigate single-neuron activity in this study, studying these specific activity bands in circuits that involve prefrontal projection populations may provide more insight into the origin of these deficits. The various projection neuron populations within the PFC can provide a combinatorial activity pattern that drives cognitive behavior and attention.

## Methods

**Lead contacts and material availability**. Further information and requests for resources and reagents should be directed to and will be fulfilled by the Lead Contact, Huibert D. Mansvelder (h.d.mansvelder@vu.nl).

*Animals*. A total of 172 rats (Charles River, Den Bosch, The Netherlands; Janvier, Le Genest-Saint-Isle, France, control groups were vendor matched) were used across all experiments (overview in Table 1). For neuroanatomical tracing experiments, and ex-vivo electrophysiological validation, 29 male Long Evans rats (8 weeks old) were housed in pairs with food and water available ad libitum. For chemogenetic experiments, 84 male Long Evans rats (8 weeks old) were initially housed in pairs with food and water available ad libitum one to two weeks before surgeries, after which they were separated for training and testing in CombiCages[23]. Rats were housed under a 12 h light/dark cycle (lights off at 12 p.m). For fiber photometry experiments, 29 male Long Evans rats were housed in pairs until surgery. After surgery for these experiments, animals were housed individually in CombiCages until finishing the testing protocol. For electrophysiology experiments, 26 male Long Evans rats were used, which underwent surgery at 8 weeks of age, and were then housed in pairs until the start of the experiment. All experimental procedures were in accordance with European and Dutch law and approved by the central committee animal experiments and local animal ethical care committee of the VU University and VU University Medical Center (Amsterdam, Netherlands).

*Viral vectors and tracers*. For anterograde tracing of dorsal and ventral mPFC projections, we infused AAV2-CaMKIIα-eYFP (UPenn, USA, 0.483 µl, $4 \times 10^{12}$ particles/ml). We used Red Retrobeads (0.138 µl, Lumafluor, USA) to anatomically label projection neurons in the mPFC. To retrogradely express Cre-recombinase in prefrontal projection neurons, CAV-2-Cre (IGMM, France) was infused in either DMS/VMS (0.483 µl, $1.25 \times 10^{12}$ particles/ml) or in MDL/MDM (0.345 µl, $5 \times 10^{12}$ particles /ml). For double labeling of projection populations, additional infusions with AAV-retro-EF1a-FLPo (0.483 µl, $1.25 \times 10^{12}$ particles/ml, Addgene 55637) were performed in DMS/VMS. For double labeling with fluorophores in the mPFC, a mixture of 1 µl containing AAV5-hSyn1-dFRT-mCherry (UZH, Switzerland, $3.4 \times 10^{12}$ particles /ml) and AAV5-EF1α-DIO-eYFP (UPenn Vector Core, USA, $2.1 \times 10^{12}$ particles/ml) was infused. The DREADD-receptor hM4D(Gi) was expressed in mPFC using AAV5-EF1α-DIO-hm4D(Gi)-mCherry (UZH, Switzerland, 0.483 µl, $3.6 \times 10^{12}$ particles/ml). DREADD control animals were infused with

**Table 1 Overview of all experimental groups and number of animals.**

| Experiment | Groups | Number of animals |
|---|---|---|
| Anterograde tracing | Dorsal/ventral mPFC | 1 |
| Retrograde tracing (Retrobeads) | MDL/MDM/DMS/VMS | 6/9/3/6 |
| Double labeling | Dorsal mPFC + DMS + MDL/ventral mPFC + VMS + MDM | 2/2 |
| Ex-vivo electrophysiology | Dorsal mPFC + MDL | 4 |
| Chemogenetics (thalamus) | MDL/MDM/eYFP | 16/16 /14 |
| Chemogenetics (striatum) | DMS/VMS/eYFP | 12/14/12 |
| Fiber photometry | MDL/MDM/DMS/VMS/eYFP | 8/4/7/6/4 |
| Slice electrophysiology | MDL/MDM/DMS/VMS | 7/6/6/7 |
| Total | | 172 |

AAV5-EF1α-DIO-eYFP (UPenn Vector, $4.2 \times 10^{12}$ particles/ml). For fiber photometry, we unilaterally expressed GCaMP6m in the mPFC using AAV5-CAG-FLEX-GCaMP6m (UPenn Vector core, 0.483 µl, $4.7 \times 1012$ particles/ml). Fiber photometry control animals were infused with AAV5-EF1α-DIO-eYFP (UPenn Vector Core, 0.483 µl, $4.7 \times 1012$ particles/ml). For slice electrophysiology experiments, we unilaterally injected 278 nl AAV9-Syn-Chronos-GFP-WPRE-bGH (UPenn Vector Core, $1.13 \times 10^{13}$ particles/ml) in the dorsal or ventral mPFC for DMS/VMS targeting and a mixture (~1:4, retrobead:virus) of red retrobeads and AAV9-Syn-Chronos-GFP-WPRE-bGH in the same dorsal and ventral mPFC locations for MDL/MDM experiments. Animals were not tested within three weeks of virus injection to allow for sufficient expression.

*Surgery.* For all experiments, rats were anesthetized with 2.5% isoflurane gas mixed with air and oxygen and delivered with a flow rate of 1.2 L/min. The rats were placed on a heating pad in a stereotaxic frame (Kopf, USA) and their skin of the scalp was retracted to expose the skull. A craniotomy was made at the location stated below and the virus/Retrobead infusion was done using a Nanoject II (Drummond Scientific, USA) via a glass micropipette. After the infusion, we held the pipette in place for 8 min to allow for diffusion, retracted it for 100 µm, waited 1 min, repeated this procedure, and then slowly retracted the pipette to minimize virus/Retrobead leakage. The following infusion coordinates (from bregma), under a 10° angle unless otherwise indicated. DMS: Anteroposterior (AP): +1.44 mm; Mediolateral (ML): +/−2.78 mm, Dorsoventral (DV): −4.47 mm. VMS: (AP + 1.44 mm, ML +/− 2.59 mm, DV 7.41 mm + 6.80 mm). MDL: (AP −3 mm, ML +/− 2.32 mm, DV 5.89 mm). MDM: (AP −3.00 mm, ML 1.42 mm, DV 5.89 mm). Dorsal mPFC: (AP + 2.76 mm, ML +/− 1.30 mm, DV −2.90 mm). Ventral mPFC: (AP + 2.76 mm, ML +/− 1.47 mm, DV: −4.87 mm. Slice electrophysiology in MD and striatum at a 0° angle: Dorsal mPFC: (AP + 2.76 mm, ML +/− 0.70 mm, DV −3.10 mm), Ventral mPFC: (AP + 2.76 mm, ML +/− 0.50 mm, DV: −5.10 mm). As indicated above in the VMS two infusions at different DV locations were made to cover the dorsal-ventral extent of this target region. Red Retrobeads and calcium indicators were infused unilaterally, whereas all other virus infusions were performed bilaterally. For the fiber photometry experiments, we implanted the fiber optic cannulas (pre-assembled from Doric lenses, NA 0.51, core diameter 400 µm, fiber length 4.5 mm for dmPFC targets, 5.5 mm for vmPFC targets) directly after, at the same location as the virus infusions. In addition, we attached stainless steel screws (0.7 mm diameter, Jeveka) to the skull to improve head cap stability. Fibers were fixed to the skull using UV-cured dental cement (RelyX, 3 M). To minimize suffering from surgeries, as an analgesic, Rimadyl (carprofen, 5 mg/kg), was administered a day before the surgery, on the day of the surgery, and two days afterward. Also, the analgesic temgesic (buprenorphine, 0.05 mg/kg) was administered once, 30–60 min before the surgery. During surgeries, lidocaine (xylocaine) was used as a local anesthetic. Immediately after the surgery, before waking up, animals received 1 ml 0.9% saline.

*Histology and immunofluorescence.* Rats were anesthetized with Euthasol (AST Farma, The Netherlands) and perfused transcardially, first with 200 ml 0.9% saline followed by 300 ml of 4% paraformaldehyde. Brains were removed and kept in the same fixative for 24 h and were then transferred to PBS with 0.02% NaN₃. Coronal sections of 50 µm were cut on a vibratome. Sections from the Retrobead experiments were directly mounted on glass slides using 2% Mowiol. Immunofluorescent stainings were performed for either NeuN, mCherry, GAD-67, or GFP. We used the following antibodies: mouse anti-NeuN (Abcam, 1:1000) with Alexa Fluor 647 donkey anti-mouse (Thermo Fisher Scientific, 1:400), rabbit anti-RFP (Rockland, 1:1000) with Alexa Fluor 546 donkey anti-rabbit (Thermo Fisher Scientific, 1:400), mouse anti-GAD-67 (Millipore, 1:1000) with Alexa Fluor 647 donkey anti-mouse (Thermo Fisher Scientific, 1:400), and rabbit anti-GFP (Abcam, 1:1000) with Alexa Fluor 488 donkey anti-rabbit (Thermo Fisher Scientific, 1:400). The sections were washed and permeabilized in PBS with 0.25% Triton X before being incubated for 3 h with blocking solution containing PBS, 0.3% Triton X and 5% normal goat serum. Next, sections were incubated overnight with primary antibodies in blocking solution at 4 °C. The following day, the sections were rinsed with PBS and incubated with secondary antibodies in a blocking solution for 2 h at room temperature. Images were acquired with a Nikon Eclipse Ti confocal microscope.

**Acute brain slice preparation.** Coronal slices of rat MD or striatum were prepared for electrophysiological recordings. Rats (4–6 months old) were anesthetized (5% isoflurane, i.p. injection of 0.1 ml/g pentobarbital) and perfused with ice-cold N-Methyl-D-glucamin (NMDG) solution containing (in mM): NMDG 93, KCl 2.5, NaH2PO4 1.2, NaHCO3 30, HEPES 20, Glucose 25, sodium ascorbate 5, sodium pyruvate 3, MgSO472H2O 10, CaCl2*2H2O 0.5, at pH 7.3 adjusted with 10 M HCl. Brains were removed and incubated in ice-cold NMDG solution. MD or striatum brain slices (250 µm thick) were cut in ice-cold NMDG solution and subsequently incubated for 15–30 min in 34 °C. Before the start of experiments, slices were allowed to recover for at least 1 h at room temperature in carbogenated (95% O2/5% CO2) ACSF solution containing (in mM): NaCl 120, KCl 2.5, NaH2PO4 1.4, NaHCO3 25, Glucose 21, sodium ascorbate 0.4, sodium pyruvate 2, CaCl2*2H2O 2, MgCl*6H2O 1 24. All recordings were made between 31.1 °C and 33.6 °C.

**Electrophysiology.** After obtaining a stable giga seal, a ramp current was injected from 0 to 500pA to assess baseline rheobase current. Spike frequency was determined both by increasing steps of current injection and by constant suprathreshold current injection. ACSF with 10 µm CNO was washed in for at least 5 min before rheobase current and spike frequency were determined again.

For voltage-clamp and current-clamp experiments, borosilicate glass patch-pipettes (3–5 MΩ, resulting in access resistances typically between 7 and 12 MΩ) were used with a K-gluconate-based internal solution containing (in mM): K-gluconate 135, NaCl 4, MgATP 2, Phosphocreatine 10, GTP (sodium salt) 0.3, EGTA 0.2, HEPES 10 at a pH of 7.4. Reciprocally connected MD neurons were targeted using the somatic expression of red retrobeads and striatal medium spiny neurons were targeted based on morphology. Data was sampled using a Multiclamp 700 B amplifier (Axon Instruments) and pClamp software (Molecular Devices) at 20 kHz and low-pass filtered at 2 kHz. Neurons were filled with 2–4% biocytin for reconstruction.

Chronos-induced postsynaptic currents (PSCs) were recorded in voltage-clamp at −60 mV. Chronos was activated by blue light (470 nm, 10 sweeps, 10 Hz, 5 pulses of 1 ms) using a DC4100 4-channel LED-driver (Thorlabs, Newton, NJ) as a light source. The light source was directed as far away from the soma as possible (typically > 200 µm) and the illumination area was limited using a diaphragm such that reliable but minimal activation was achieved. Light intensity was adjusted to elicit a half maximum amplitude (typically > 10 pA) of the first EPSC to prevent overstimulation of the axon boutons (Collins et al., 2018)[24].

*SP-5-CSRTT task.* Behavioral experiments were done in modified, self-paced, and automated 5-CSRTT environments. We constructed CombiCages by connecting a macrolon home-cage to an operant chamber (Med Associates Inc., St. Albans, VT, USA) using a custom-made polymer tube with a diameter of 10 cm. Operant chambers were equipped with five cue holes containing LED stimulus lights and infrared beam detectors on one side. A food magazine, a red magazine light, and a yellow house light were positioned on the opposite wall. This setup significantly reduces the number of days required to train an animal to baseline 5-CSRTT performance. Animals can initiate trials during a 2.5-h window every day, and can perform hundreds of trials during a session, while still meeting performance thresholds normally used in conventional 5-CSRTT experiments. In some cases, animals can earn enough food to maintain their daily caloric intake, thereby avoiding the need for food restriction. Validation of this paradigm can be found in Bruinsma et al.[23].

We placed the rats in the CombiCages[23] two days before the training in the task started. During training, animals earned their food in the form of precision pellets in the task (Dustless Precision Pellets, grain-based, F0165, 45 mg, Bio-Serve, USA). To maintain the rats' weight to an 85–90% food restriction regime, we provided additional standard food chow.

Acquisition of SP-5-CSRTT performance was established by different training phases. First, animals learned to associate pellet delivery with reward. In this phase, for 50 trials a pellet was delivered after a variable delay. A reward was signaled by the magazine light, and a magazine response started the next trial. In the subsequent phase, rats needed to nose poke in one of five illuminated cue holes to

earn a reward for 50 trials. Next, only one of the 5 cue holes was illuminated and responses into this hole after a delay of 5 s led to rewarding delivery. During this phase, incorrect or premature nose pokes were not punished. Animals needed to complete 100 trials in this stage.

In the final phase, the animals needed to respond to the cue after a fixed delay of 5 s. The cue hole was lit for a specific cue duration which was initially 16 s and was reduced to 1 s in five steps. The rats had to nose poke during the cue within a 2 s limited hold period after cue presentation. A lack of response was considered an omission and resulted in a timeout period of 5 s. Premature responses, nose pokes during the delay, or incorrect responses were also punished with a 5 s timeout period. Correct responses were always rewarded with a pellet.

After a correct response, animals could start the next trial 5 s after the reward collection of the pellet. Importantly, animals could only initiate during the first 2.5 h of the dark cycle[23]. In this final phase, the performance criterion to reach the following stage with shorter SD was a minimum of 50 started trials, accuracy (ratio of correct and incorrect responses, see below) >80%, and either omission <20% or correct trials >200 in the current stage. The program monitored these parameters online using a sliding window of 20 trials. If rats passed the performance criterion, the program automatically moved to the next shorter cue duration[23].

Chemogenetic inactivation was performed in cognitively challenging sessions in which either the delay was randomly varied between 5, 7.5, or 12.5 s to test inhibitory control, or sessions in which the cue duration was varied between 0.2, 0.5, or 1 s to test attentional aspects of the task[27].

Fiber photometry sessions were performed in similarly cognitively challenging sessions. In addition, rats were also retrained to baseline performance in an operant cage without homecage attachment, which was more suited to tethered recordings.

*Drug administration.* Two weeks before testing, animals were habituated to injections by giving them several saline injections. Directly before a testing session, Clozapine N-oxide (CNO) dihydrochloride (Hello Bio, UK) was dissolved in 0.9% saline and injected intraperitoneally (i.p.) 30 min prior to the start of the dark phase. Solutions were freshly prepared on each test day and doses were administered using a Latin square design. Animals received either 0, 5, or 10 mg/kg CNO per testing session, in randomized order, based on recent work in rats[25].

**Fiber photometry.** Rats used for fiber photometry were trained in CombiCages until baseline performance and then recorded for 4–6 sessions, each lasting up to 150 min. We used a setup based on the one used by Lerner et al.[58]. (Fig. S5b for a schematic overview), centered around a lock-in amplifier (RZ5P, Tucker-Davis Technologies, USA) that controls two excitation LEDs (405 nm at 531 Hz, and 490 nm at 211 Hz; Thor Labs M490F1 and M405F1). This setup allowed us to use the isosbestic wavelength of GFP as a control for motion-induced and other systemic noise since the 405 nm channel will contain all incoming signals except specifically GCaMP-emission. The light was then led through a filter cube (FMC4 AE(405)_E (460–490)_F(500–550)_S, Doric Lenses) into the fiber optic rotary joint. Rats were tethered to the recording setup with a patch cord (MFP_400/440/LWMJ-0.53.FC-ZF2.5, Doric Lenses) and a fiber optic rotary joint (FRJ_1 × 1_FC-FC, Doric Lenses). Emitted light from GCaMP6m was led back to the filter cube into a photodetector (Newport Femtowatt 2151), which then transmitted signal back to the lock-in amplifier which demodulated both incoming channels into separate signal traces. Data was then recorded on a dedicated recording PC using Synapse (Tucker-Davis Technologies). Incoming behavioral signals were also transmitted from the operant chamber to the lock-in amplifier using a MedPC SuperPort card (DIG-726, MedAssociates) and corresponding cable (CMF, Tucker-Davis Technologies). Using this system, we could reliably perform chronic recording experiments for over 3 months.

**Exclusion criteria.** Twelve animals with a misplaced virus or retrobead infusions were excluded (MDL:3, MDM: 6, DMS: 0, VMS: 3), as were 15 rats for the chemogenetic experiments that had unilateral virus expression or that did not establish stable baseline performance (MDL: 5, MDM: 5, eYFP (MD): 1, DMS: 2, VMS: 2, eYFP (Str): 0). In addition, for the photometry experiments, 15 rats with misplaced fibers or no virus expression were excluded (i.e., no GCaMP6m-positive neurons in the tissue volume that allows successful capture of emission, as found in Fig. S5i–j; MDL: 2, MDM: 6, DMS: 3, VMS: 4, GFP: 0). For slice electrophysiology experiments, three outliers were removed, one had a capacitance above 500 pF, and two had a $R_{input}$ above 340 MΩ, exclusion did not affect the outcome.

*Cellular quantification.* For the Retrobead experiments, maximum intensity Z projections of 5 z-planes were made using ImageJ. Next, images were overlayed with a rat brain atlas at AP + 3.00 mm, +2.76 mm, or +2.52 mm. Subregions of the mPFC were included as ROIs. Layers of the PFC were determined using the Swanson brain atlas and were validated with NeuN sections. Cells were counted manually using ImageJ per ROI and the area of the ROIs was determined. For the double-labeling experiments, composite images were created for signals from eYFP, GAD-67, and mCherry. Cells were counted manually. For the DREADD experiments, the areas of virus expression were selected as an ROI in ImageJ. The area of the ROI was calculated and cells within the ROI were counted manually.

*Chemogenetics and behavioral analysis.* Behavioral data were acquired with MED-PC software (Med-Associated, USA). All data analyses and statistics were done with custom-written scripts in MATLAB (Mathworks, USA). We calculated the percentage accuracy as: #correct/ (#correct + #incorrect) * 100. Premature responses and omissions were expressed as a percentage of the total number of trials. All latencies were expressed in seconds. Trials with a magazine latency >10 s were excluded from further analysis[23]. The normality of the data was tested with the Shapiro-Wilk test. Time-dependent effects of CNO were analyzed by splitting the 2.5 h session into five blocks of 30 min. Two-way mixed repeated measures ANOVAs were employed with time and dose as within-subject factors[23]. To compare the effects of CNO in the different projection groups, three-way mixed repeated-measured ANOVAs were employed with dose and delay or cue duration as within-subject factors and the group as the between-subjects factors. Additional parameters, such as the number of started trials, were not dependent on delay or cue duration and effects of CNO were tested with two-way mixed repeated-measures ANOVAs with dose as within-subject factor and group as between-subject factor. Post hoc testing was done using Wilcoxon rank-sum tests or *t*-tests with Benjamin-Hochberg false discovery rate (FDR) to adjust *p* values for multiple comparisons. For the neuroanatomical data, a Chi-Square independence test was used to test differences in mediolateral distributions between projection populations in dorsal and ventral mPFC. In the ex-vivo electrophysiological experiments, a non-parametric Mann–Whitney-*U* test was used to assess the effects of CNO on mCherry (putative DREADD)-positive cells versus control neurons. To test the effects of CNO on the distribution of premature responses, Friedman tests were performed between the doses, and *p*-values were corrected for multiple testing. In all cases, the significance level was set at $p < 0.05$. Data are presented as mean +/− SEM throughout the main text and figures and as mean +/− SD in the supplementary tables.

**Fiber photometry analysis.** Fiber photometry data were analyzed using custom-made MATLAB scripts. In short, raw data from the TDT RZ5P recording system was first corrected for motion and other systemic noise by fitting the 405 nm-channel to the 470nm-channel and dividing, resulting in a raw δF/F (F being the adjusted 405 nm-channel). We then lowpass filtered the signal on 1 Hz and highpass filtered on 30 Hz. We then performed a spectral analysis to correct for the remaining low-frequency noise. Finally, we down-sampled the signal by a factor of 64, yielding a final frame rate of around 16 Hz, which was our final δF/F. For all subsequent analyses, we used small time windows around the trial. To be able to standardize signals and look only for changes in population activity associated with the task, we aligned every trace to a baseline period between −5 and −1 s before the start of each trial. Since we included a 10 s inter-trial interval after each trial where rats could not initiate a new trial, the baseline should not include any trial-related signals. To test differences in signal between delay periods, we only looked at the signal between trial initiation and the cue presentation time of the longest delay (12.5 s). We either used the Friedman test (comparison between trial outcomes within the group) or Kruskal–Wallis test (comparison between groups), with post hoc uncorrected Dunn's tests and Benjamini–Hochberg's false discovery rate to adjust *p* values. Significance for ANOVAs was set on $p < 0.05$. To assess the difference from the baseline, we calculated bootstrapped confidence intervals with 5000 iterations and an alpha of 0.001. In short, we randomly sampled mean signal traces for each outcome type for each rat and took the mean of each random sample (each random sample being the same as the total number of rats in the group), and repeated 5000 times. We then took a confidence interval with an alpha of 0.01 of all 5000 mean traces of a given trial outcome, yielding an interval between the 99.9th and 0.01st percentile value for each data frame, which we considered as boundaries between which the signal could be. We then took averages of the upper and lower confidence interval bounds of all rats to construct the group confidence interval. To study differences between signal traces of two experimental groups or two outcomes, we performed permutation tests that compared distributions at every data point. For each data point, we considered the distributions significantly different if the alpha was < 0.01. For both the bootstrapping and permutation tests, singleton significant points (i.e., data points with no neighbors that were also significant) were filtered out of the data set. One data frame corresponded to approximately 125 ms.

**Electrophysiology analysis.** Chronos-evoked PSCs were calculated by taking the median over 10 sweeps that were corrected for drift using a robust regression fit. Paired-pulse-ratios were calculated by dividing the peak of $PSC_N$ by $PSC_1$. Chronos-evoked PSC latency was calculated as time to reach 80% of peak value from the light onset. Input resistance was calculated using the slope of the linear fit to the current-voltage curve using negative current steps between 0 and −100 pA (15 or 20 pA increments, 0.5 or 1 s duration), using the steady-state voltage in the last 200 ms of the step. The membrane time constant tau determined by the median overfitting a first-order exponential function (only goodness of fit >0.8 used) to the first 300 ms to the voltage trace in response to three negative current steps between 0 and 50 pA (15 or 20 pA increments, 0.5 or 1 s duration). Capacitance was calculated as input resistance over membrane time constant. Sag was calculated as the percentage difference between the Δ peak voltage and Δ steady-state (last 1/5th of the step duration) from baseline in response to a

negative step current (0.5 or 1 s) that elicited a Δ peak voltage closest to −20 mV. Burst and steady-state firing frequency were calculated based on the number APs (threshold at 0 mV) in the 50 ms after the first AP (burst) or the last 200 ms (steady-state) of positive current steps between 0 and 200 pA (50 pA increments, 0.5 s duration). Some neurons were recorded with 15 pA increments, here steps with less than 5pA difference from the 50 pA increments steps were used. Biocytin-filled neurons were reconstructed in Neuromantic software (V1.6.3) and plotted for illustrative purposes using the Neuroanatomy toolbox in ImageJ. Offline data analysis was performed in Graphpad Prism 6 and Matlab 2019a. No assumptions were made about the data distribution and all analyses were done using non-parametric Friedman test with post hoc Dunn's and Benjamini-Hochberg's false discovery rate corrected Mann-Whitney U-tests for repeated measures and Mann–Whitney U-tests for simple comparisons, significance set at $P < 0.05$.

**Reporting summary**. Further information on research design is available in the Nature Research Reporting Summary linked to this article.

## Data availability
All raw datasets are available upon reasonable request. Source data are provided with this paper.

## Code availability
Code is available on the public Github repository: https://github.com/sybrendekloet/mPFCprojections [59]

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

## Acknowledgements
We thank H. Lodder, R. Elsinga, M. van der Roest, and T. Brouwer for technical support, and A. J. Timmerman for support with animal caretaking.

## Author contributions
Conceptualization: S.F.d.K., B.B., H.T., R.M., T.P., and H.D.M.; investigation: S.F.d.K., B.B., H.T., T.S.H., E.M.J.P., A.R.v.d.B., and A.L..; data analysis: S.d.K., H.T., and B.B.; writing–original draft: S.d.K.; writing–review and editing: B.B., H.T., R.M., T.P., and H.D.M..; funding acquisition: H.D.M.; resources: T.P. and H.D.M.; supervision: R.M., T.P., and H.D.M.

## Disclosures

HDM received funding for this work from the Netherlands Organization for Scientific Research (NWO; VICI grant 865.13.002), European Union's Horizon 2020 Framework Programme for Research and Innovation under the Specific Grant Agreement No. 785907 (Human Brain Project SGA2), and NWO Gravitation program BRAINSCAPES: A Roadmap from Neurogenetics to Neurobiology (NWO: 024.004.012).

**Competing interests**
The authors declare no competing interests.
