## [Peer Review File · Nature Communications]

Reviewers' Comments:

Reviewer #1:

Remarks to the Author:

De Kloet et al., present a tour-de-force analysis of mPFC complexity during the classic 5CSSRT task. This work is technically impressive in combining circuit-specific techniques (DREADD / photometry), and conceptually impressive because it illuminates the incredible specificity of the medial prefrontal cortex, which to my mind has not been shown before in task with this complexity. In general, the paper is a fundamental advance.

I had several questions that might help me understand this work better:

- 1) It was challenging for me to keep track of the experiments. Would it clearer to have a circuit-specific summary at the beginning or the end, and to revisit this summary in each figure? Would it be clearer to term the manipulation dmPFC ->MDM and vmPFC->MDL, dmPFC->DMS and vmPFC->VMS?
- 2) Would information on sample size and statistical tests be clearer in the results section and the figure legends? Are there measures of effect size?
- 3) I had difficulty seeing the main effects in Figure 2G-J. I think the individual plots are stacked next to the bars, but both are too small and the colors too similar to interpret. I wonder if a something like a beeswarm plot would help (as in S4A). The authors could also apply this approach to Fig 3I-L, Fig S3A-L, and Fig 1A-D. The bar graphs would be redundant with this. A line can indicate central tendency.
- 4) For injection sites, the cartoons and exemplars are quite helpful. However, not all injection sites are represented consistently throughout, as in Fig 3C
- 5) I wonder if the differences in input resistance in Fig 5 between dmPFC-MDL and vmPFC-MDM were related to laminar differences in Fig 1A/B (more L5 for dmPFC-MDL?). I wonder if the overall story might be clearer if Fig 5 followed Fig 1. I also tried to compare the laminar distribution with figures in Gabbott et al., (ref #9) – perhaps a few sentences in the discussion might clarify similarities / differences?
- 6) There may be readers interested in the data in Fig S3G-H. First, it appears that the saline baseline is quite different for each condition (i.e., G/K vs. I,H,J,L). Second, I wonder if the ks-test is the best test here – perhaps the fixed-interval timing literature (start times, curvature, efficiency, or some other metric) can be helpful. Looking at the data, I'm not sure I can conclude that 'no effect of CNO on the temporal distribution of premature response latencies'
- 7) How does this literature link to cognitive control? Also, some groups have linked rodent prefrontal activity to 4 Hz cognitive-control rhythms that explicitly malfunction in schizophrenia/Parkinson's disease (see PMID: 27989675 or 28348382 for rodent-> human links, PMID 24835663 for a review, and Figure 5 of PMID 30528064). Although the authors don't measure neuronal signals at this time resolution, how would they speculate the dmPFC->MDM signals integrate with this larger literature? Making these connections would likely greatly expand the impact of this work to humans and to human disease.

Minor:

- 1) Absolute numbers (i.e., X #s of cells) in the results section (Page 5) might help contextualize these percentages.
- 2) When a p-value is presented, the test statistic and test might help interpret the results on Page 5 – and throughout.
- 3) There appear to be multiple cue lengths? I lost track of how this was studied in the manuscript.
- 4) The approach to significant figures might be checked. For instance, on page 43, $F=8$, and then later $F=0.92$. There are a few other examples of this.
- 5) I found it hard to identify patterns from Table S1 /S2. Would a heatmap or hinton diagram be clearer?
- 6) Table S3 – is green text difference then green boxes in prior tables?
- 7) Figure 4C legend: – what is a <0.01 ? Is this a multiple comparisons correction?
- 8) Page 23 – 'Superficial' – does the anatomy reveal very many superficial prefrontal neurons?

9) More detail on CombiCages might be useful.

10) How many rats were not included for what reasons across all experiments? The exclusion criteria section seems incomplete.

Reviewer #2:

Remarks to the Author:

In this work de Kloet et al study the role of different mPFC projections , into different parts of the mediodorsal thalamus (the MDM/MDL) and the medial striatum (DMS/ VMS), in controlling the goal directed responses of rats in 5-choice serial reaction time test. The authors find that the mPFC neurons projecting into the different sites differ in location, in function and in the downstream neurons they innervate. Very interestingly they identify that inhibiting the projections into the two different parts of the MD create an opposite effect on premature responses in prolonged delay time suggesting differential role in driving responsive action, the fact that inhibiting the MDL not only reduced premature responding but also increased omissions, according to the authors suggest a role in driving response. The MDM according to the authors could play a role in withholding response. A similar role is suggested to the mPFC-DMS projections which like the mPFC-MDM create more premature responses when inhibited. The Authors support their claims using fiber photometry with which they record the activity of the differently projecting mPFC populations during the task and compare the profiles of activation as predictors of the specific behavioral outcome. The work seems to be based on a vast amount of work and the results seem solid.

I really appreciate the approach of taking different populations of neurons from the same region according to their projection site and to the differential circuit they are involved in and measure the activity in the same task, especially in the mPFC which is indicated to be involved in so many aspects of behavior. Here the authors take the aspect of behavioral control which is a highly studied mPFC trait and map it onto different circuits going through the mPFC. I think this kind of work is highly important to the field of system and circuit neuroscience. I do believe this paper should be accepted and suitable for publication in nature communications; however, I do have some minor concerns and suggestions:

a. I do think the paper could benefit from sharpening of the story. The paper is very rich with results yet on several occasions I did not find how the paper relates them to a general theme or story.

Examples: 1. Fig 3 I-L presents the differences between the activity of the differently projecting neurons on several parameters such as kinetics, timing and peaks , in several of those figures MDL and DMS show similar patterns, how does it relate to the previous result where in behavior it seems that it is the MDM and the DLS projections seem to work in the same direction? How is this division between dorsal and ventral projection neurons relates to the role they are playing? 2. Figure 5 is a very nicely done e-phys experiment with some interesting results but I couldn't find where does the story relates to it - how does the different traits of the different downstream neurons relate to the role of the projection in the behavior? What does the paired pulse facilitation in the MD vs the no change in the DMS or depression in the vms mean in relation to the story of the paper? In relation to their role in behavioral control?

b. The fact that there are four different projections, and four different downstream regions is what makes this paper so interesting yet it also makes it hard to follow, therefore I think the authors should make it as clear as possible – for example the whole color coding in figure 3 is hard to follow where orange could be DMS or premature, and red could be strong activation or incorrect or DMS. Some figures such as figure 4 are too small – the text but also the effect – noticing and identifying the two colors making that line (or sometime dot) is tough.

c. In figures 2d and 3d – what is the data set? Is this pooled data? Example figure?

Point-by-point reply to Reviewer comments

Reviewer #1

De Kloet et al., present a tour-de-force analysis of mPFC complexity during the classic 5CSRT task. This work is technically impressive in combining circuit-specific techniques (DREADD / photometry), and conceptually impressive because it illuminates the incredible specificity of the medial prefrontal cortex, which to my mind has not been shown before in task with this complexity. In general, the paper is a fundamental advance.

I had several questions that might help me understand this work better:

Reviewer #1, point 1: It was challenging for me to keep track of the experiments. Would it be clearer to have a circuit-specific summary at the beginning or the end, and to revisit this summary in each figure? Would it be clearer to term the manipulation dmPFC →MDM and vmPFC→MDL, dmPFC→DMS and vmPFC→VMS?

Author Reply: This is a valid point. The different experimental lines and projection populations and amount of results necessitate a clear structure and we should assist readers as much as possible. To this end, we have made a summary figure, which we have added to the manuscript as Figure 6. This figure is described and referred to in the discussion, and it provides a graphical summary of all experiments.

As suggested by the reviewer, we have also edited figures 3, 4, 5, and S5 (former Figures 2, 3, 4 and S4), so that the groups are no longer called 'MDL', 'DMS', etc. but rather 'dmPFC-MDL', 'dmPFC-DMS'. We also changed e.g. dmPFC-MDL in the text to dmPFC→MDL in many instances. We hope this makes it a bit clearer.

Reviewer #1, point 2: Would information on sample size and statistical tests be clearer in the results section and the figure legends? Are there measures of effect size?

Author Reply: We have added information about the tests used in the main text, and sample sizes are reported in the figure legends. We have also added effect sizes to our behavioral results, which should also help interpret the main effects shown in Figure 3G-J (former Figure 2G-J) that are also addressed in the next question.

Reviewer #1, point 3: I had difficulty seeing the main effects in Figure 2G-J. I think the individual plots are stacked next to the bars, but both are too small and the colors too similar to interpret. I wonder if a something like a beeswarm plot would help (as in S4A). The authors could also apply this approach to Fig 3I-L, Fig S3A-L, and Fig 1A-D. The bar graphs would be redundant with this. A line can indicate central tendency.

Author Reply: We agree that the figures can get quite dense and hard to read. Therefore, as suggested by the reviewer, we changed the bars in the new Figure 3G-J (former Fig 2), and indicated median and error bars. Also, we have increased the contrast between different categories, and we also decreased the y-axis range, so that the data spread becomes clearer. We think this should increase legibility of the figure.

We did a similar thing for the new Figure 4I-L (former Fig 3), and Figure S4A, M-P (former Fig 3), so that individual data points are more legible and interpretable. We changed Figure 1A-D in a slightly different way, so that the individual data points stand out more clearly.

Reviewer #1, point 4: For injection sites, the cartoons and exemplars are quite helpful. However, not all injection sites are represented consistently throughout, as in Fig 3C

Author Reply: We agree that an overview figure can be quite insightful. Therefore, we have added a similar figure for our DREADD expression, it is now visible in the new Figure 3B (former Figure 2). This should help interpret the coverage of our chemogenetic manipulations.

Reviewer #1, point 5a: I wonder if the differences in input resistance in Fig 5 between dmPFC-MDL and vmPFC-MDM were related to laminar differences in Fig 1A/B (more L5 for dmPFC-MDL?).

Author Reply: This is a very interesting angle. Laminar differences may also reflect distinct innervation into MDL and MDM by driving and modulatory inputs. We have added our thoughts on this in the discussion in the second paragraph on page 27 (blue text).

Reviewer #1, point 5b: I wonder if the overall story might be clearer if Fig 5 followed Fig 1.

Author Reply: We agree with the reviewer and in line with the suggestion, we changed former Fig 5 into the new Figure 2.

Reviewer #1, point 5c: I also tried to compare the laminar distribution with figures in Gabbott et al., (ref #9) – perhaps a few sentences in the discussion might clarify similarities / differences?

Author Reply: As suggested by the reviewer, we have now expanded the discussion of the link between our anatomical data and the data presented in the elaborate study by Gabbott et al. While our findings mostly agree, there could be some differences in layer distribution of dmPFC-DMS projecting neurons. However, the injections by Gabbott covered a much larger portion of the DMS, so it is difficult to compare. We have included our thoughts on this into our discussion on page 24 (blue text).

Reviewer #1, point 6) There may be readers interested in the data in Fig S3G-H. First, it appears that the saline baseline is quite different for each condition (i.e., G/K vs. I,H,J,L). Second, I wonder if the ks-test is the best test here – perhaps the fixed-interval timing literature (start times, curvature, efficiency, or some other metric) can be helpful. Looking at the data, I'm not sure I can conclude that "no effect of CNO on the temporal distribution of premature response latencies"

Author Reply: We agree with the reviewer, the ks-test is indeed not sufficient to state that CNO has no effect on the temporal distribution of premature response latencies. We have now statistically evaluated proportionalized data using a Friedman test, which compares the distributions of each condition. We found that the Friedman test was not significant for any

condition, which contribute more to the claim that premature response distribution is not affected by CNO. We've changed Supplementary Figure 4G-L (Former Figure S3G-L) so that it shows proportions of responses, rather than the absolute number, and have added the test results in the figure legend.

Reviewer #1, point 7) How does this literature link to cognitive control? Also, some groups have linked rodent prefrontal activity to 4 Hz cognitive-control rhythms that explicitly malfunction in schizophrenia/Parkinson's disease (see PMID: 27989675 or 28348382 for rodent-> human links, PMID 24835663 for a review, and Figure 5 of PMID 30528064). Although the authors don't measure neuronal signals at this time resolution, how would they speculate the dmPFC->MDM signals integrate with this larger literature? Making these connections would likely greatly expand the impact of this work to humans and to human disease.

Author Reply: We appreciate these suggestions by the reviewer, there could absolutely be merit in looking into the link between our findings and clinical implications. We have added this point of view into the final paragraph of our discussion on page 30 (blue text).

Minor:

1) Absolute numbers (i.e., X #s of cells) in the results section (Page 5) might help contextualize these percentages.

Author Reply: We have added absolute numbers for each percentage we mention in the text.

2) When a p-value is presented, the test statistic and test might help interpret the results on Page 5 – and throughout.

Author Reply: We went through the manuscript, and have added test statistics and test information to each statistical test used.

3) There appear to be multiple cue lengths? I lost track of how this was studied in the manuscript.

Author Reply: We have clarified our use of multiple cue lengths, which are used in separate test sessions from the variable delay periods. We have added this in the chemogenetics part of our results section. We have also edited Figure 3C (former Figure 2C), so that it now shows the cue as 1s, which is consistent with the data represented in the rest of the figure.

4) The approach to significant figures might be checked. For instance, on page 43, $F=8$, and then later $F=0.92$. There are a few other examples of this.

Author Reply: We now report all test statistics with two decimal digits, so that it is consistent.

5) I found it hard to identify patterns from Table S1 /S2. Would a heatmap or hinton diagram be clearer?

Author Reply: We agree with the reviewer. We tried to make heatmaps or hinton diagrams for the table data, but in the end, we were not satisfied by the clarity of these figures either. Hence, we restructured the table and made it as clear as possible, with descriptions in the table legend arranged in a more accessible way.

6) Table S3 – is green text difference then green boxes in prior tables?

Autor Reply: See our answer to minor point 5: We agree that the layout of the tables was confusing. We tried to make heatmaps or hinton diagrams for the table data, but in the end, we were not satisfied by the clarity of these figures either. Hence, we restructured the table and made it as clear as possible, with descriptions in the table legend arranged in a more accessible way.

7) Figure 4C legend: – what is $\alpha < 0.01$? Is this a multiple comparisons correction?

Autor Reply: This indicated a parameter used for the permutation test. We have added some additional information about the test parameters into the results section that accompanies Figure 5 (former Figure 4), with a reference to the Methods section, where this is explained into greater detail.

8) Page 23 – ‘Superficial’ – does the anatomy reveal very many superficial prefrontal neurons?

Autor Reply: We agree that it is not the best way to describe our findings and changed this part of the discussion so that it better reflects our data, which suggest that the ‘superficial’ projection neurons are really located primarily in layers 2, 3 and 5, which is indeed not really superficial.

9) More detail on CombiCages might be useful.

Autor Reply: Details on the CombiCages appear in a previous publication from our lab (Bruinsma et al., 2019). However, to assist the reader, we now added a few sentences about the technique into the results section, where we also refer to a more extensive explanation in the Methods section (page 34, “SP-5-CSRTT task” section, first paragraph, blue text). This should clarify the experimental setup, and how it provides significant improvements in number of trials started, while retaining similar behavioral performance to conventional 5-CSRTT approaches.

10) How many rats were not included for what reasons across all experiments? The exclusion criteria section seems incomplete.

Autor Reply: We have added the number of rats that were excluded for each experiment, it can now be found in the same paragraph in the Methods section as the exclusion criteria (page 36-37, “Exclusion criteria”).

Reviewer #2

In this work de Kloet et al study the role of different mPFC projections, into different parts of the mediodorsal thalamus (the MDM/MDL) and the medial striatum (DMS/ VMS), in controlling the goal directed responses of rats in 5-choice serial reaction time test. The authors find that the mPFC neurons projecting into the different sites differ in location, in function and in the downstream neurons they innervate. Very interestingly they identify that inhibiting the projections into the two different parts of the MD create an opposite effect on premature responses in prolonged delay time suggesting differential role in driving responsive action, the fact that inhibiting the MDL not only reduced premature responding but also increased omissions, according to the authors suggest a role in driving response. The MDM according to the authors could play a role in withholding response. A similar role

is suggested to the mPFC-DMS projections which like the mPFC-MDM create more premature responses when inhibited. The Authors support their claims using fiber photometry with which they record the activity of the differently projecting mPFC populations during the task and compare the profiles of activation as predictors of the specific behavioral outcome. The work seems to be based on a vast amount of work and the results seem solid. I really appreciate the approach of taking different populations of neurons from the same region according to their projection site and to the differential circuit they are involved in and measure the activity in the same task, especially in the mPFC which is indicated to be involved in so many aspects of behavior. Here the authors take the aspect of behavioral control which is a highly studied mPFC trait and map it onto different circuits going through the mPFC. I think this kind of work is highly important to the field of system and circuit neuroscience. I do believe this paper should be accepted and suitable for publication in nature communications; however, I do have some minor concerns and suggestions:

Reviewer #2, point a: I do think the paper could benefit from sharpening of the story. The paper is very rich with results yet on several occasions I did not find how the paper relates them to a general theme or story.

Examples: 1. Fig 3 I-L presents the differences between the activity of the differently projecting neurons on several parameters such as kinetics, timing and peaks, in several of those figures MDL and DMS show similar patterns, how does it relate to the previous result where in behavior it seems that it is the MDM and the DLS projections seem to work in the same direction?

How is this division between dorsal and ventral projection neurons relates to the role they are playing?

Autor Reply: This is a very interesting point. We hypothesize that each projection population is part of a distinct neural network that guides distinct types of behavior. However, we agree that this does not emerge clearly in our discussion. Therefore, we have added several parts into our discussion to further clarify that each projection stands on its own, and that even though their activity profiles may seem similar, the different projection targets can be part of distinct functional networks. For instance, the DMS acts as a striatal input, and prefrontal input may serve as a bias to guiding striatal output into a particular direction – in this case, we suggest that it may well be a “stop” signal (see also Ardid 2019, or Cox & Witten 2019 for striatal function review). The MDL, however, is part of distinct circuitry, and is integrated in a dmPFC-MDL recurrent loop circuit, which keeps dmPFC neurons activated throughout a delay (Collins et al., 2018, Parnaudeau et al., 2019). This circuit may be part of a system that underlies response readiness and guides a more action-oriented behavioral response, i.e. a “go” signal. One way this could possibly be regulated is through collaterals of MDL-projections that target other thalamic nuclei, that are more integrated in motor circuits (Collins et al., 2018). We have added these distinctions into the discussion. Additionally, we have emphasized the distinct roles of each projection population in different parts of the discussion, and have added a summary figure that gives a visual representation of the circuitry as we interpret it. Hence, we think that due to their positioning in distinct (and larger) brain networks, these different prefrontal projection populations can have distinct roles in behavior, even though they have similar activity profiles.

We have now added this to the Discussion section, on page 27 for MDL/MDM (blue text), and on 28-29 for DMS (blue text).

Reviewer #2, point a continued, example 2. Figure 5 is a very nicely done e-phys experiment with some interesting results but I couldn't find where does the story relates to it - how does the different traits of the different downstream neurons relate to the role of the projection in the behavior?

What does the paired pulse facilitation in the MD vs the no change in the DMS or depression in the vms mean in relation to the story of the paper? In relation to their role in behavioral control?

Autor Reply: This is a valid point, and in combination with reviewer 1's point 5b, we have now moved former Figure 5 to follow Figure 1, so that it ties in better with the anatomical data.

We think there are several points to take away from this data. First, the differences in postsynaptic responses show that prefrontal input may not only be distinct at the somatic level, but is also handled differently in target regions, giving rise to further specialization of each pathway. Second, we think that each pathway represents a distinct brain network, and this divergence in processing could underlie this distinction because prefrontal signals are transmitted to downstream targets in different ways. Third, we show for the first time that processing within the MD occurs in distinct ways depending on which subregion of the MD is targeted. Taken together, these points support the notion that these pathways are operating concurrently and are driving different networks.

Reviewer #2, point b. The fact that there are four different projections, and four different downstream regions is what makes this paper so interesting yet it also makes it hard to follow, therefore I think the authors should make it as clear as possible – for example the whole color coding in figure 3 is hard to follow where orange could be DMS or premature, and red could be strong activation or incorrect or DMS. Some figures such as figure 4 are too small – the text but also the effect – noticing and identifying the two colors making that line (or sometime dot) is tough.

Autor Reply: We agree that the amount of data can be overwhelming and hard to follow. We have therefore done a slight overhaul of most figures, and have added a summary figure (Figure 6) in the discussion section that should encompass every result and our interpretation.

In new Figure 3 (former Fig 2), we have changed the colour coding, so that premature responses no longer have the same colour as DMS-projections. Also, we have changed new Figure 4 (former Fig 3) so that the bars are wider, and the text is more legible.

Reviewer #2, point c. In figures 2d and 3d – what is the data set? Is this pooled data? Example figure?

Autor Reply: These are example traces of one single rat, during one session. To state this more clearly in the figure legend, we added an extra line to clarify this.

Reviewers' Comments:

Reviewer #1:

Remarks to the Author:

I read the authors response and the revised manuscript. I find it considerably improved and I have no further comments. This paper is a technical tour-de-force and is likely to have a major impact on the field.

Reviewer #2:

Remarks to the Author:

I have no further concerns - all were answered.